# A Cat Is A Cat (Not A Dog!): Unraveling Information Mix-ups in Text-to-Image Encoders through Causal Analysis and Embedding Optimization

**Chieh-Yun Chen**🏃‍♂️🔺 **Chiang Tseng**🏃‍♂️ **Li-Wu Tsao**🏃‍♂️ **Hong-Han Shuai**🏃‍♂️

🏃‍♂️ National Yang Ming Chiao Tung University 🔺Georgia Institute of Technology

🔺cchen859@gatech.edu 🏃‍♂️{chiang.ee11,lwtsao.ee09,hhshuai}@nycu.edu.tw

## Abstract

This paper analyzes the impact of causal manner in the text encoder of text-to-image (T2I) diffusion models, which can lead to information bias and loss. Previous works have focused on addressing the issues through the denoising process. However, there is no research discussing how text embedding contributes to T2I models, especially when generating more than one object. In this paper, we share a comprehensive analysis of text embedding: i) how text embedding contributes to the generated images and ii) why information gets lost and biases towards the first-mentioned object. Accordingly, we propose a simple but effective text embedding balance optimization method, which is training-free, with an improvement of 125.42% on information balance in stable diffusion. Furthermore, we propose a new automatic evaluation metric that quantifies information loss more accurately than existing methods, achieving 81% concordance with human assessments. This metric effectively measures the presence and accuracy of objects, addressing the limitations of current distribution scores like CLIP's text-image similarities. The code is available: https://github.com/basiclab/Unraveling-Information-Mix-ups.

## 1 Introduction

Text-to-image (T2I) diffusion models [11, 12, 14, 17, 18] have recently captured significant attention. Subsequent research [4, 9, 13, 16, 1, 20, 3] has extensively explored the roles of cross-attention and self-attention mechanisms in the denoising process to enhance image control. Despite these advancements, there remains a critical gap in understanding the role of text embedding within T2I models, particularly in scenarios involving the generation of multiple objects. For instance, Fig. 1 illustrates that when prompted with "a lion and an elephant," models like Stable Diffusion often generate an ambiguous creature that blends features of both (or object missing on the right), highlighting issues of semantic interpretation and token embedding. This paper delves into how text embeddings in-

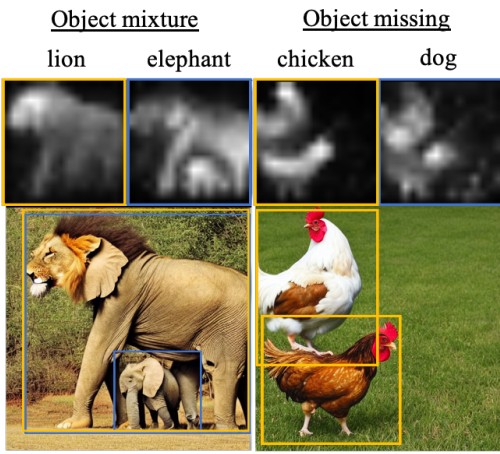

Object mixture / Object missing

lion / elephant / chicken / dog

*a lion and an elephant* / *a chicken and a dog*

Figure 1: Visualization of cross-attention maps when object mixture and missing occur.

fluence the semantic outcomes of generated images, identifying specific problems of information bias and loss due to the causal nature of the self-attention mechanism in text encoders.

38th Conference on Neural Information Processing Systems (NeurIPS 2024).

The existing literature [1, 3, 16, 20, 13] has worked in image latents to address information loss, e.g., objects missing; however, there is no research on the main cause of the problem——text embedding. Therefore, this paper focuses on text embedding to investigate the issue. We first analyze the information bias in text embedding and find that the generated objects tend to bias towards the first mentioned object, as Table 1 shows, when the given prompt contains more than one object. The reason for biasing towards the first mentioned object is due to the causal manner in which making $n^{th}$ token embedding contains the weighted attention of token embeddings between 0 to $(n-1)^{th}$ tokens, as illustrated in the right-bottom side of the Fig. 2. Moreover, we analyze the semantic contribution of the token embeddings by masking different tokens. Table 2 and Fig. 3 demonstrate that the causal manner would contribute to the accumulation of general information in the special token, e.g., end of token (<eot>), and padding token (<pad>). Masking the embeddings of given tokens still allows the T2I model to generate the expected information using the remaining special tokens' embeddings.

Due to the significant information loss issues in T2I models, such as object mixing and missing, we dissect the generative process of the T2I model to pinpoint the origins of these losses. In the text embedding, the causal manner would make the embedding of the $n^{th}$ token mixed with the token embedding between 0 to $(n-1)^{th}$, which makes the later token embedding similar to the earlier embedding. In the case of a prompt "a cat and a dog", the causal manner would mix the <dog> embedding with the <cat> embedding. This similarity in embeddings results in similar distributions on cross-attention maps, as detailed in Sec.4.4. When denoising directions on these maps align too closely throughout the denoising steps, it can cause a mixed representation if responses are equally low, or one object may overshadow the other if its map elicits a stronger response, leading to object disappearance. These phenomena are visualized in Fig. 1. To address the issue of information bias and loss, we propose the Text Embedding Balance Optimization (TEBOpt) to promote distinctiveness between embeddings of equally important objects for preventing mixing and working alongside existing image latent optimization techniques to address object disappearance. The main contributions of this paper are outlined as follows:

- This paper examines how text embedding contributes to generated images in text-to-image diffusion models and demystifies how the causal manner leads to information bias and loss while contributing to general information.

- We propose the Text Embedding Balance Optimization solution containing one positive and one negative loss to optimize text embedding for tackling information bias with 125.42% improvement in stable diffusion.

- We propose an evaluation metric to measure information loss. Compared to the CLIP score for evaluating text-image similarity, and the CLIP-BLIP score for evaluating text-text similarity, our evaluation metric provides a concrete number for identifying whether the specified object exists in the generated image.

## 2   Preliminaries

Text-to-image diffusion models [11, 12, 14, 17, 18] typically contain the text encoder [15, 6], a variational autoencoder (VAE), and a denoising UNet, as Fig. 2 demonstrates. Given a text prompt, the text encoder would first obtain the text hidden states $h_s \in \mathbb{R}^{N \times D}$ from the sum of token embedding and positional embedding, where $N$ is the maximum token length in the text encoder and $D$ represents the embedding dimension; then, it calculates the text embedding by going through the encoder layers with self-attention mechanism and causal masking manner. Next, given the text embedding $\varepsilon$ and the initial image noise $z_t$, the denoising UNet $\epsilon_\theta(z_t, \varepsilon, t)$ would gradually denoise latents in each timestep $t$ to get the final image by iteratively predicting the noise residuals conditioned on the text embedding and the previous denoised latents.

Notably, the causal masking manner in the text encoder makes every token have information only from its previous tokens, which causes the text embedding to have information bias. The bottom of Fig. 2 illustrates how the causal manner works in the self-attention mechanism [21]. The query $\boldsymbol{Q}$, key $\boldsymbol{K}$, and value $\boldsymbol{V}$ are calculated as follows:

$$\boldsymbol{Q} = \ell_Q(h_s), \quad \boldsymbol{K} = \ell_K(h_s), \quad \boldsymbol{V} = \ell_V(h_s), \quad \text{where} \quad \boldsymbol{K}, \boldsymbol{Q}, \boldsymbol{V} \in \mathbb{R}^{h_n \times N \times D/h_n} \quad (1)$$

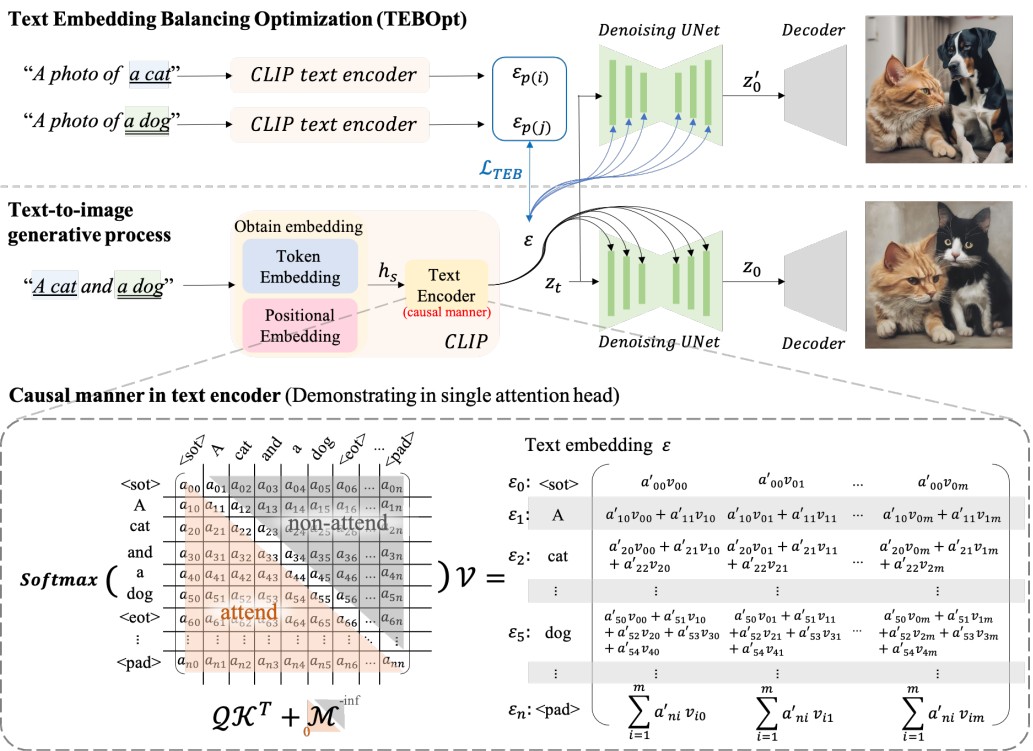

Figure 2: Overview of the text-to-image generative model, including the details of the causal manner in attention mechanism. Because of the causal nature of the embedding, information is accumulated from the starting token through the end of the sequence, resulting in bias in the earlier token. To balance the critical information, we propose text embedding optimization for purifying the object token with equal weights within their corresponding embedding dimension.

$$\text{Attention}(\boldsymbol{Q}, \boldsymbol{K}, \boldsymbol{V}) = \text{softmax}\left(\frac{\boldsymbol{Q}\boldsymbol{K}^\top + \mathbf{M}}{\sqrt{d_k}}\right)\boldsymbol{V}, \quad \mathbf{M}_{ij} = \begin{cases} 0, & \text{if } j \leq i \\ -\infty, & \text{if } j > i \end{cases}, \quad (2)$$

where $\ell_Q$, $\ell_K$, and $\ell_V$ are learned linear projections, and $h_n$ represents the head number. $d_k$ is the dimension of $\boldsymbol{K}$ and $\mathbf{M}$ is the causal mask. Upon receiving the attention weight $\boldsymbol{Q}\boldsymbol{K}^\top \in \mathbb{R}^{N \times N}$, the mechanism applies the causal mask $\mathbf{M}$. Then, taking $\boldsymbol{V}$ as a weight metric to get the weighted sum of attentions, representing the embedding for each token. Due to the causal manner, the embedding of the $n^{th}$ token ($\varepsilon_n$) contains weighted information for the tokens 0 to $n-1$. This representation causes two issues: First, the earlier token information accumulates the most since every subsequent token has it. Furthermore, the later token's embedding is not a pure representation of itself, potentially causing identity loss. For example, given a text prompt "a cat and a dog", the earlier token <cat>'s information would be mixed into the embedding of the later token <dog>. It makes the generated image have a higher possibility of generating two cats, as the middle row of Fig. 2 shows.

# 3 Analysis and method

In this section, we analyze how causal manner affects the text embedding as well as the generated images in text-to-image diffusion models.

## 3.1 Information bias in the text embedding

We investigated 400 different prompts with different random seeds with structures (a) "a/an <object1> and a/an <object2>" and (b) "a/an <object2> and a/an <object1>" by exchanging the position of <object1> and <object2> in prompt (a), where objects are randomly sampled from 17 different animals.

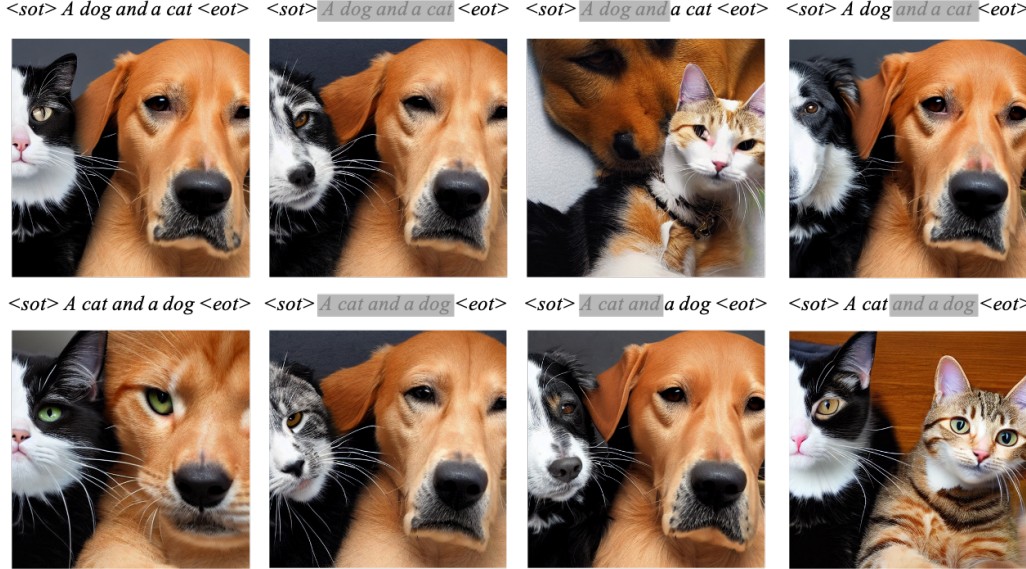

Figure 3: Masking text embedding to identify the contribution of critical tokens, e.g., cat/dog, and special tokens, e.g., <sot>, <eot>, <pad>. The first row and the second row both contain cat and dog inside prompt but in different order. The analysis shows that special tokens contain general information about the given prompt. However, the cat/dog tokens carry more weight than the special tokens. In the last two columns, where one of the animal token embeddings is masked while retaining the special tokens' embedding, the generated image is predominantly influenced by the remaining animal's token embedding.

Table 1 shows that within the same set of two animals and the same random seed, both bias on the earlier mentioned animal. In prompt (a), the statistics with pink background show that it causes 67.5% missing issue, including 46.0% generating only object 1. Similarly, in prompt (b), it causes 69.5% missing issue, including 47.0% generating only object 2. The information bias (Info bias) is defined by $\frac{\text{\# of only obj1 exist}}{\text{\# of only obj2 exist}}$ and, in both prompts, they are 2.30 and 0.46 respectively, which are far from balanced (Info bias = 1).

| Prompt | (a) A/An <obj1> and a/an <obj2> | (b) A/An <obj2> and a/an <obj1> |
|---|---|---|
| 2 objects exist | 12.25% | 11.75% |
| mixtures | 20.25% | 18.75% |
| only obj1 exist | 46.00% | 21.75% |
| only obj2 exist | 20.00% | 47.00% |
| no target object | 1.50% | 0.75% |
| Info bias | 2.30 | 0.46 |

Table 1: Both prompts strongly bias towards the first mentioned object. The bias generally exists in more objects, reported in Supplement D.

### 3.1.1 Causal effect for generating images with more than one object

Driven by the observation of information bias, we investigated the text embedding contribution of critical tokens. In Stable Diffusion, we first utilize CLIP [15] as the text encoder to obtain the text embedding $\varepsilon_i \in \mathbb{R}^{N \times D}$, where $i \in [0, N-1]$. Before the denoising UNet gets the text embedding and image latent, we mask the chosen embedding and generated the corresponding images. Table 2 provides the quantitative analysis in 400 different prompts with the same setting of (a) used in Sec. 3.1, extending with three masking settings corresponding to Fig. 3's each column.

Firstly, the default experiment in Table. 2 demonstrate stable diffusion has 20.25% mixture, 67.50% missing ratio, and 2.3x information bias to the first mentioned object. Regarding the visualization sample of the default experiment, the first column in Fig. 3 shows that animals (cat and dog) within one prompt in different orders with the same initial image latent would generate different results. The bottom one demonstrates the mixture issue, making the right animal simultaneously similar to dog and cat. Regarding the second experiment, we mask all the embeddings of the given tokens, where $\varepsilon_i = -inf, i \in [1, 5]$. It reduces 2.5% mixture rate but increases 11.75% missing rate. With only

one object existing, the existing rate for object 1 and object 2 is more balanced as the embeddings of all given tokens are masked. It suggests that the remaining embeddings, which are special tokens, including <sot>, <eot>, and <pad>, contain the information of the masked embeddings due to causal manner in the text encoder.

An important insight here is that eliminating dominant embeddings reduces the mixture rate. One of the causes of the mixture issue is the attention mixture, as the text embedding in Fig. 2 shows. Given a prompt "a cat and a dog", as the text embedding is projected as key and value multiplied with the query to get cross-attention maps during denoising, the <cat> embedding would trigger the <cat> response while the <dog> embedding would trigger both <dog> and <cat> responses, causing animal mixtures. Once one of the object's responses dominates, the other animal disappears, leading to a missing issue. In the third and fourth columns, we mask one of

| Analysis | Default | Mask 1-5 | Mask 1-3 | Mask 3-5 |
|---|---|---|---|---|
| 2 objects exist | 12.25% | 3.00% | 2.75% | 0.50% |
| only mixture | 10.00% | 11.50% | 5.25% | 3.50% |
| obj1 + mixture | 6.50% | 2.75% | 0.00% | 0.75% |
| obj2 + mixture | 3.75% | 3.50% | 3.75% | 0.00% |
| mixture sum | 20.25% | 17.75% | 9.00% | 4.25% |
| only obj1 exist | 46.00% | 33.25% | 3.00% | 89.00% |
| only obj2 exist | 20.00% | 42.00% | 82.00% | 1.75% |
| no target objects | 1.50% | 4.00% | 3.25% | 4.50% |
| missing sum | 67.50% | 79.25% | 88.25% | 95.25% |
| Info bias | 2.30 | 0.79 | 0.04 | 50.86 |

Table 2: **Analysis of masking token embeddings.** Masking all the given token would reduce the mixture issue but increase the missing issue with balanced object 1 and object 2 existing rate. Masking one of the objects would not completely eliminate the masked object's information but would significantly reduce its existing rate. The implementation details are in Supplement B.

the animals' text embeddings, resulting in generating only the remaining animal. These experiments suggest that, firstly, causal manner makes the information accumulate from earlier token to later token. Even masking critical text embeddings, special tokens, i.e., <sot>, <eot>, <pad>, can generate the given prompt information. In addition, although special tokens accumulate information to form general embeddings, when the text embeddings contains critical tokens' embeddings, the critical token would lead the generated results, as the last two columns show.

### 3.1.2 De-causal effect for generating images with more than one object

**Hypothesis.** *Replacing the token embedding of later mentioned object from the corresponding pure embedding, the hypothesis expects to solve the bias problem and information loss.*

Following the setting of the prompt structure (a) in Sec. 3.1, with a text prompt "a/an <obj1> and a/an <obj2>", the token index of <obj1> and <obj2> are taken into the critical token set $O = \{2, 5\}$. It first generates the pure embedding of the later mentioned one, "a photo of a <obj2>" ($\varepsilon_{p_{obj2}}$). Then, replace the corresponding token embedding of <obj2> in the original embedding ($\varepsilon$) with the pure one. The combined text embedding ($\varepsilon'$) is as follows:

$$\varepsilon' = \begin{cases} \varepsilon_i & , \text{if } i \notin O \quad or \quad i = \min(O) \\ \varepsilon_{p_{obj_n},5} & , else \end{cases}, \qquad (3)$$

| Prompt | Default | Hypothesis |
|---|---|---|
| 2 objects exist | 12.25% | 7.00% |
| only mixture | 10.00% | 10.50% |
| obj1 + mixture | 6.50% | 3.25% |
| obj2 + mixture | 3.75% | 3.50% |
| only obj1 exist | 46.00% | 44.00% |
| only obj2 exist | 20.00% | 30.25% |
| no target object | 1.50% | 1.50% |
| Info bias | 2.30 | 1.45 |

Table 3: **Analysis of Hypothesis.** Replacing the token embedding of later mentioned object from the corresponding pure embedding can balance the information but lead to a large drop of two objects coexistence.

where n refers to the $n^{th}$ object. Table 3 reflects that directly replacing original embedding with the pure embedding would balance the information of object 1 and 2; however, it would result in a 5.25% loss in 2 objects coexistence. A nearly equal probability of generating only objects 1 and 2 proves that replacing token embedding with pure embedding eliminates accumulated information about object 1.

### 3.2 Method: balancing critical information in the text embedding

While the causal manner results in information bias and information loss, it contributes to generate image content aligned with the general prompt information. To eliminate the accumulated information

but retain general information, we design a Text Embedding Balancing Optimization, called TEBOpt (uppermost in Fig. 2), cooperating with image latent optimization to address these issues. The TEBOpt tackles information bias and object mixture issues while the latent optimization tackles object mixture and missing issues. Since the missing object is caused by an insufficient response value and an inadequately activated region in the cross-attention map corresponding to that object, and the text embedding is unable to precisely determine the activated position, our method cooperates with existing latent optimization methods to address this issue.

Regarding a prompt containing two objects, we expect their corresponding text embedding to be unmixed instead of mixing earlier tokens' embeddings. Since the ultimate goal is to preserve the general information of the two objects, we cannot directly replace the original embedding with the pure object's text embedding. This would result in a high probability of losing one of the objects. Thus, our proposed TEBOpt contains a TEB loss in order to encourage the later mentioned token's embedding to be less similar to the earlier token's embedding, while at the same time being as similar to its pure embedding. Considering a text prompt with $k$ objects in a set $O$, we first obtain each text embeddings $\varepsilon_p \in \mathbb{R}^{N \times D}$ and take the critical token embedding $\varepsilon_{p,i} \in \mathbb{R}^{1 \times D}$ as the pure embedding. For example, the prompt "a dog and a cat" contains two objects and the pure prompt embedding is calculated in a format prompt of ["a photo of a <dog>", "a photo of a <cat>"]. In summary, the TEB loss is as follows:

$$\mathcal{L}_{TEB}^{pos}(\varepsilon, \varepsilon_p) = \min_{i \in O} sim(\varepsilon_i, \varepsilon_{p(i)}), \tag{4}$$

$$\mathcal{L}_{TEB}^{neg}(\varepsilon) = \frac{1}{k(m-1)} \sum_{i \in O} \sum_{\substack{j=1 \\ j \neq i}}^{m-1} sim(\varepsilon_i, \varepsilon_j), \tag{5}$$

$$\mathcal{L}_{TEB} = -\mathcal{L}_{TEB}^{pos} + \mathcal{L}_{TEB}^{neg}, \tag{6}$$

where $sim(u, v) = \frac{\mathbf{u}_i \cdot \mathbf{v}_i}{\|\mathbf{u}_i\| \|\mathbf{v}_i\|}$ and m means the effective token count, where we do not include <sot> and <eot> for optimization. The implementation details are included in Supplement B. After the text embedding is optimized, the image latent would be updated conditioned on the loss design for cross-attention maps during the denoising process. For example, A&E [1] contains a loss function to ensure that each selected token activates some image patches in the cross-attention map. SynGen [16] designs a loss function to encourage the cross-attention map of the relative token to be similar and make the cross-attention map of the unrelative token dissimilar.

## 4 Experiment

### 4.1 Experimental settings

**Baselines.** We compare our proposed method with the default Stable Diffusion 1.4 [17], and 3 state-of-the-art (SOTA) baselines, including Structure Diffusion [3], Attend-and-Excite [1], and SynGen [16]. All of them focus on improving attribute bindings or solving object missing in text-to-image diffusion models. However, there is no one considering the information bias. In this literature, the objective is to analyze information balance caused by pretrained text encoders instead of surpassing existing SOTAs on solving object missing. Therefore, we would provide the experimental results of our proposed method on top of the baselines.

**Data.** We follow previous methods [1, 20] to create a set of 400 prompts with the format "a/an <obj1> and a/an <obj2>" with corresponding random seeds. The objects are sampled from 17 different animals defined by previous methods [1, 20], including cat, dog, bird, bear, lion, horse, elephant, monkey, frog, turtle, rabbit, mouse, panda, zebra, gorilla, penguin and chicken.

### 4.2 Evaluation metrics

To analyse the issues of mixed objects and missing objects in generated images, we designed an automated evaluation method since existing metrics, e.g., text-image similarity using CLIP [15] or text-text similarity using CLIP and BLIP [7], used in SOTAs [1, 20] cannot provide the exact counting number to indicate whether the object exists or not. Detailed discussions are provided in Supplement

C. First, we employed a pre-trained object detection model, OWL-ViT [10], which is a SOTA in open-vocabulary object detection. We separate the text prompt into k objects and the model separately predicts bounding boxes and confidence scores for the corresponding objects. Mixture status is determined when the overlap of the two bounding boxes for different objects exceeds 90%. Here, we use two different thresholds to detect mixture objects and single objects, ensuring detection accuracy. Additionally, to validate the effectiveness of this automatic metric, we conducted a human evaluation to demonstrate that its results are highly correlated with human perception. We asked users to label 400 generated images, each categorized into one of five options: i) two objects exist, ii) mixture exists, iii) missing object 1, iv) missing object 2, or v) no objects exist. Our automatic evaluation metric achieves an accuracy of 81% based on human responses, demonstrating its effectiveness.

## 4.3 Qualitative results

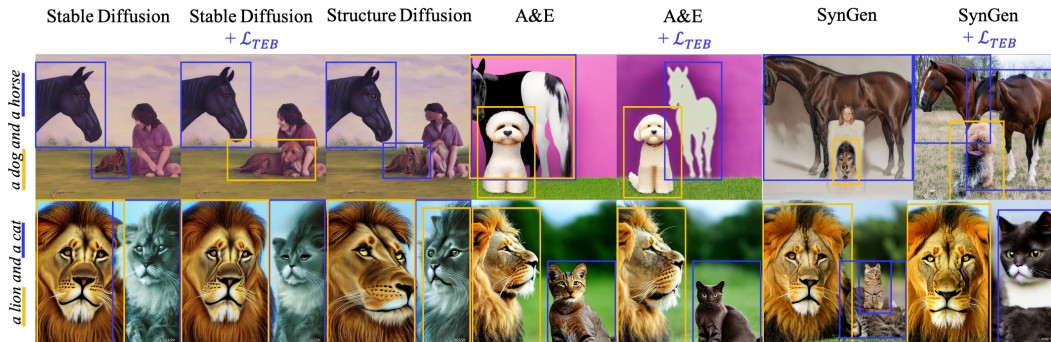

Figure 4: Qualitative comparison of all methods. Every prompt uses the same seed.

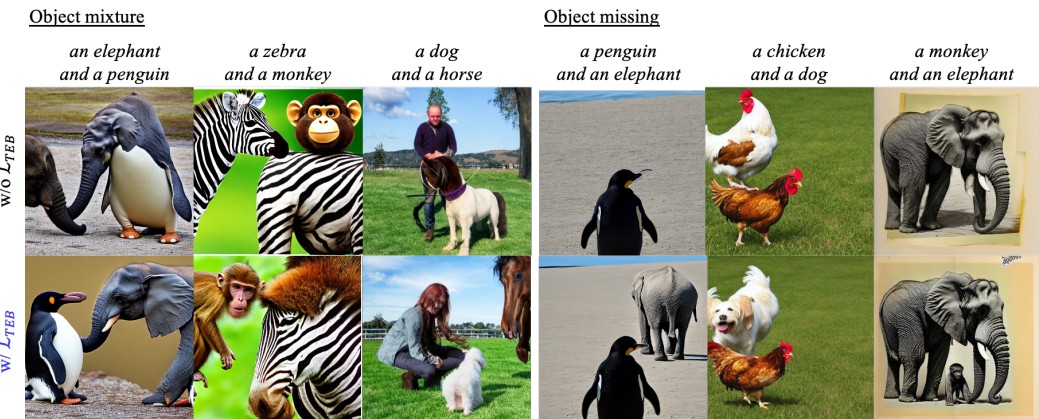

Figure 5: Qualitative comparison for the generated image with vs. without $\mathcal{L}_{TEB}$ in Stable Diffusion 1.4. Every prompt uses the same seed.

Fig. 4 demonstrated the visual comparison between all methods. The same color of the bounding box and the underline for the prompt indicate the same object. The bounding boxes are predicted based on our proposed evaluation method discussed in Sec. 4.2. As an object is wrapped by two bounding boxes with different colors, e.g., A&E [1] in the first row or Structure Diffusion [3] in the second row, it has a high potential to contain mixed objects. In the first row in Fig. 4, our TEBOpt ($\mathcal{L}_{TEB}$) helps Stable Diffusion generate the specified dog and help A&E to make the horse dissimilar from the dog. In the second row, our TEBOpt ($\mathcal{L}_{TEB}$) helps Stable Diffusion generate a cat less similar to a lion, which is initially equipped with a mane as a visual signal of a lion. Regarding Fig. 5, we demonstrate the generated image with and without our proposed TEBOpt ($\mathcal{L}_{TEB}$). Using our TEBOpt, T2I models solved both the object mixture and missing issues, especially when trying to solve the problem of object mixtures. It is worth noting that objects mixture or missing are also affected by the denoising process in T2I models. Our text optimization mainly contributes to balancing the information in the text embedding and further reducing mixed and missing issue.

## 4.4 Quantitative results

Table 4 provides the comparative experiment for object mixture, missing and information bias. As Structure Diffusion [3] manipulates key and value from text embeddings in denoising UNet's cross-attention calculation, which works on text embedding as ours, our method working on top of Stable Diffusion can directly compare with it.

Among all methods with our TEBOpt ($\mathcal{L}_{TEB}$) can further improve the objects' existing balance with 125.42%[1] in Stable Diffusion, 78.43% in A&E, and 10.65% in SynGen. Though the balance of Structure Diffusion is better than ours, it causes a 7.5% decrease in the probability of 2 objects coexisting. Since it directly combines pure text embeddings and original embedding, it loses general information with two objects in the text embedding, as the Sec. 3.1.2 discusses.

| Method | Stable Diffusion | $+\mathcal{L}_{TEB}$ | A&E | $+\mathcal{L}_{TEB}$ | SynGen | $+\mathcal{L}_{TEB}$ | Structure Diffusion |
|---|---|---|---|---|---|---|---|
| 2 objects exist | 14.5% | +0.5% | 34.0% | +1.0% | 52.0% | +4.5% | 7.0% |
| only mixture | 9.5% | -1.0% | 11.5% | -4.0% | 10.5% | -1.0% | 11.5% |
| obj1 + mixture | 6.5% | -1.5% | 9.0% | -1.5% | 6.5% | -3.5% | 4.0% |
| obj2 + mixture | 5.5% | -2.0% | 6.0% | +0.5% | 2.5% | +1.5% | 3.0% |
| only obj1 exist | 45.0% | -6.0% | 26.0% | -2.5% | 10.5% | -0.0% | 39.0% |
| only obj2 exist | 17.0% | +11.0% | 12.0% | +5.0% | 17.0% | -2.5% | 34.5% |
| no target objs | 2.0% | -1.0% | 1.5% | +1.5% | 1.0% | +1.0% | 1.0% |
| Info bias | 2.65 | **1.39** | 2.17 | **1.38** | 0.62 | **0.72** | 1.13 |

Table 4: **Analysis of balancing performance**. Within the cases generating one object, we highlight the better balanced results in blue and red.

Furthermore, SynGen [16] generates a reverse trend between object 1 and 2 since it works to make the two objects' cross-attention maps' distance as far as possible, which would contribute to separating two objects leading to a large improvement in object missing and information balance. With our text optimization, we can further solve mixture issue and making the balance better since we tackle the issue from the front of the problem in text embedding. Thus, text embedding optimization and image latent optimization reach out a good cooperation for solving information bias and loss. The generalizability of $\mathcal{L}_{TEB}$ in information bias for more than 2 objects is reported in Supplement D.

In Table 5, we evaluate the similarity of token embeddings and the distance of cross-attention maps between two objects within one prompt. Token embedding similarity (Token sim) is calculated by cosine similarity while the cross-attention map distance (Map dist) is calculated by the symmetric Kullback-Leibler divergence between two normalized cross-attention maps $M_i$ and $M_j$: $\frac{1}{2}(D_{KL}(M_i||M_j) + D_{KL}(M_j||M_i))$, where $D_{KL}(M_i||M_j) = \sum_{pixels} M_i \log(M_i/M_j)$. With our $\mathcal{L}_{TEB}$, the token embedding similarity between two objects reduces 36.98% and the cross-attention map distance increases 13.62%.

| Token sim ↓ | SD 1.4 | $+\mathcal{L}_{TEB}$ | Map dist ↑ | SD 1.4 | $+\mathcal{L}_{TEB}$ |
|---|---|---|---|---|---|
| Average | 0.454 | 0.286 (-36.98%) | Average | 2.321 | 2.637 (+13.62%) |
| $[\min, \max]$ | [0.291, 0.669] | [0.137, 0.537] | $[\min, \max]$ | [0.287, 5.722] | [0.553, 6.340] |

Table 5: Analysis of optimized token embedding similarity (Token sim) and cross-attention map distance (Map dist) between two objects within one prompt.

**Discussion of how the similarity of text embedding affects cross-attention maps' distance.**

We calculated the cosine similarity between various text embeddings and displayed the results in Fig. 6 (a). The data indicates that objects with similar colors or sizes, e.g., penguin-panda or turtle-frog, tend to exhibit higher similarity in their text embeddings, which can be attributed to the training mechanism of CLIP. Additionally, we computed the distance between the cross-attention maps, using

---

[1]The info bias value indicates less bias when it is closer to 1, different from the intuitive assumption that lower values represent less bias. Here's the detailed calculation: First, we calculate the bias distance between the info bias value and the balanced value (which is 1). The bias distances of SD and SD + TEBOpt are (2.647-1)/1 = 164.71% and (1.393-1)/1 = 39.29%. Note that we have rounded the values for information bias in Table 4, reporting 2.647 as 2.65 and 1.393 as 1.39. Secondly, the balance improvement is then calculated as 164.71%-39.29% = 125.42%.

the same function in Table 5, generated by the two objects' text embeddings with the same initial latent in the early denoising steps. As shown in Fig. 6 (b), there is a positive correlation between the similarity of text embeddings and the distance of the cross-attention maps triggered by the two objects. Specifically, objects with similar text embeddings are more likely to activate overlapping areas during the denoising process. This confirms that similar text embeddings contribute to object mixture, while the short distance of cross-attention maps leading to object missing has been proven by SynGen [16].

## 5 Related works

**Object mixture or missing.** Stable Diffusion [14] pointed out that stable diffusion models have the issue of concept bleeding, which occurs by unintended merging or overlap of distinct visual elements, leading to object mixture or missing. Also, the root cause lies in the usage of pretrained text encoders, including CLIP [15] and OpenCLIP [6]. However, all the existing methods investigate the issue in the denoising process instead of text encoders. For instance, Attend-and-Excite [1] proposed an optimization process to ensure every selected token triggers some image patches when calculating the cross-attention maps between text embedding and image features.

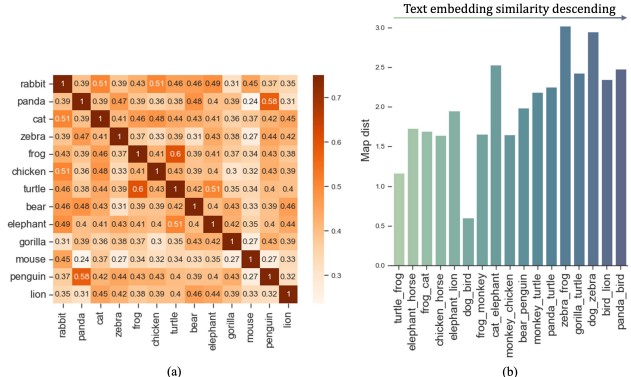

Figure 6: (a) The cosine similarity of text embedding from single word. (b) The KL distance of cross-attention maps that are triggered by two words. The data is ordered by their text embedding similarity.

SynGen [16] designed a loss function to increase cross-attention maps' similarity between modifier-entity pairs and enhance attribute binding. Structure Diffusion [3] leveraged linguistic structure to separate the given prompt into several noun phrases and modify the corresponding value to manipulate text embedding when calculating cross-attention maps in denoising steps. Predicated Diffusion [20] designed to decompose the prompt containing several objects into independent objects and minimize the loss between the generated image based on the combined independent prompt and that of the original prompt. Refocus [13] proposed to define each object's positional bounding boxes with GPT-4 and designed two loss functions to make the object only denoised inside its given region.

However, there is no task investigating the causal manner in the pre-trained text encoders. Therefore, we share a comprehensive analysis and propose a simple solution to tackle the issue of information bias, object mixture and missing, e.g., generating one animal with bear head and turtle shell when prompting "a turtle and a bear" or generating two cats when prompting "a cat and a dog".

## 6 Conclusion

In this study, we conducted a detailed analysis of text embedding's impact on text-to-image diffusion models, a topic rarely explored. Our findings indicate that the causal processing of text embedding leads to information accumulation, causing biases and loss. Directly replacing accumulated embeddings with purified embeddings, though resulting in decreased coexistence of two objects, enhances the balance between generating either object. We introduce a training-free Text Embedding Balance Optimization (TEBOpt) method that effectively eliminates problematic information in critical token embeddings, improving information balance handling in stable diffusion by 125.42% while preserving object coexistence performance. Additionally, due to the unreliability of existing metrics for assessing inaccuracies in generated images, we propose a new automatic evaluation metric to more effectively measure information loss.

# 7 Acknowledgments

This work is partially supported by the National Science and Technology Council, Taiwan under Grants NSTC-112-2221-E-A49-059-MY3 and NSTC-112-2221-E-A49-094-MY3.

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

# Supplementary Material

## A  Broader impact and safeguards

This study deepens our understanding of text embedding in text-to-image (T2I) models, showing that better insights into pre-trained text encoders can markedly enhance accurate content generation. We identify that text embeddings accumulating too much information can introduce biases and data loss, insights that are vital for developers of large vision-language models to effectively mitigate these problems. For safeguards, we have adopted the safeguard provided in Stable Diffusion [17], aligning with ethical standards and preventing misuse of generative models.

## B  Implementation details

**Computing resources and method efficiency.**  All experiments are run on one NVIDIA RTX 3090 24 GB GPU. We calculated the average inference time from 400 images. The average inference time of default Stable Diffusion 1.4 is 8.21 seconds/image while our proposed $\mathcal{L}_{TEB}$ in Stable Diffusion 1.4 is 9.14 seconds/image.

**Model architecture.**  All the analyses are conducted on Stable Diffusion 1.4, where the text encoder is the pre-trained CLIP ViT-L/14 [15] and the scheduler is the PNDMSchedular. In the text encoder, CLIP, the maximum token length (N) is 77, the embedding dimension (D) is 768, and the attention head number ($h_n$) is 12. We use a fixed guidance scale of 7.5 and set the denoising step to 50. For performing the TEB loss in Equation 6, we set a maximum optimization time as 20 and a threshold for $\mathcal{L}_{TEB}$ as -0.7, which comes from $\mathcal{L}_{TEB}^{pos} = 0.95$ and $\mathcal{L}_{TEB}^{neg} = 0.25$. The optimization target for $\mathcal{L}_{TEB}$ is -0.7 but if the optimization time exceeds 20, it will stop optimizing the text embedding. Moreover, all baselines are conducted with their official source codes.

**Cross-attention map visualization.**  We aggregate and take the average of all cross-attention maps with a size not larger than 32x32 in all 50 denoising steps. Note: when calculating the distance between cross-attention maps, we do not accumulate the cross-attention maps within 50 denoising steps. We only take the cross-attention map in corresponding denoising step.

**Masking token embedding.**  We set the selected dimension of $encoder\_attention\_mask$ in the denosing UNet as 0 while the unselected dimension is 1. The working mechanism for $encoder\_attention\_mask$ is that after key multiplying query to get the attention matrix, the dimension with 0 inside $encoder\_attention\_mask$ would add -10,000 to the attention metric while the dimension with 1 would remain unaffected. After processing the attention metric with $softmax$, the selected dimension in the attention matrix would become 0.

## C  Existing evaluation methods discussion

The existing evaluation metrics for missing objects are two-fold: text-image similarity and text-text similarity.

**Text-image similarity** is calculated by the CLIP cosine similarity between the text prompt and the corresponding generated images. A&E [1] further separated the metric into *Full Prompt Similarity* and *Minimum object Similarity*. Since full prompt similarity may not accurately reflect the existence of missing objects, minimum object similarity is proposed to evaluate the most neglected subjects independently. Take "a dog and a cat" as an example. It first separates the prompt into "a dog" and "a cat", evaluates both separately and uses the minimum score as the metric to indict missing objects.

**Text-text similarity** is calculated by the CLIP cosine similarity between the text prompt and the caption of the generated image obtained by the pretrained BLIP image-captioning model [7].

However, all these metrics cannot reflect the mixed objects issue and cannot accurately reflect how many missing objects are in the generated images. Thus, we propose a new evaluation metric for reporting the number of mixed and missing objects as discussed in Sec.4.2.

Furthermore, we discuss the 3 metrics that previous methods used, including *full prompt similarity*, *minimum object similarity* and *text-text similarity* in Figs. 7, 8 and 9. In Fig. 7, given the prompt "a cat and a penguin", the generated image on the left contains a penguin and one mixture of penguin and cat, while the generated image on the right contains a penguin and a cats. However, the left one, which contained the mixture object, has 3.57% higher in the *full prompt similarity* than the right one without mixture objects. It indicates that the *full prompt similarity* cannot properly measure the mixture cases. Moreover, the *text-text similarity* of the left one is 59.56% higher than the right one, which indicates that it cannot identify the mixture issue. In contrast, our proposed metric can accurately predict mixture/independent objects with a specific number.

**Prompt: "a cat and a penguin"**

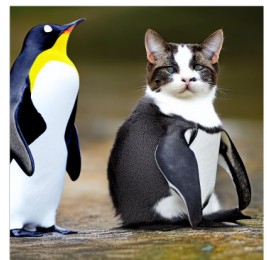

- Full prompt similarity:  0.3712                    0.3584
- Minimum Object similarity:  0.2668                  0.2723
- Text-text similarity:  0.8830                       0.5534
- Generated caption by BLIP:  a cat sitting next to a       a cat standing next to a
                              penguin on the ground    penguin on a brown background
- Our proposed metric:  1 penguin and 1 mixture       1 cats and 1 penguin

Figure 7: *Full prompt similarity* and *text-text similarity* cannot properly evaluate the mixture cases. As indicated in red color, the image on the left contains a mixture of cat and penguin but its *full prompt similarity* and *text-text similarity* are higher than which of the image on the right.

In Fig. 8, the image on the left generates a bear and a mixture of bear and cat, while the image on the right has a bear and two cats. In terms of *text-text similarity*, the left one is 8.68% higher than the right one. Again, it indicates that it cannot identify the mixture issue. In contrast, our proposed metric can accurately report that the left one contains 1 bear and 1 mixture, while the right one contains 1 bear and 2 cats.

**Prompt: "a bear and a cat"**

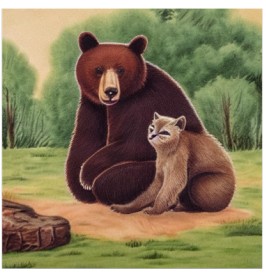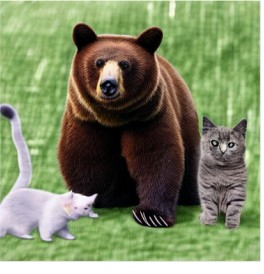

- Full prompt similarity:  0.3175                    0.3313
- Minimum Object similarity:  0.2092                  0.2617
- Text-text similarity:  0.8430                       0.7757
- Generated caption by BLIP:  a painting of a bear and       a bear, a cat, and a cat
                              a baby bear               on a grass field
- Our proposed metric:  1 bear and 1 mixture          1 bear and 2 cats

Figure 8: *Text-text similarity* of the left one is 8.68% higher than that of the right one. It indicates that the metric cannot identify the mixture issue.

In Fig. 9, both images generate a mixture of bear and lion but the left one has a 33.48% higher *text-text similarity* than the right one. In terms of *full prompt similarity* and *minimum object similarity*, they have a difference of 7.01% and 15.27% respectively. These variations make the evaluation metrics unreliable to measure object mixture and missing. In contrast, our proposed metric can accurately report that both images contain one mixed object.

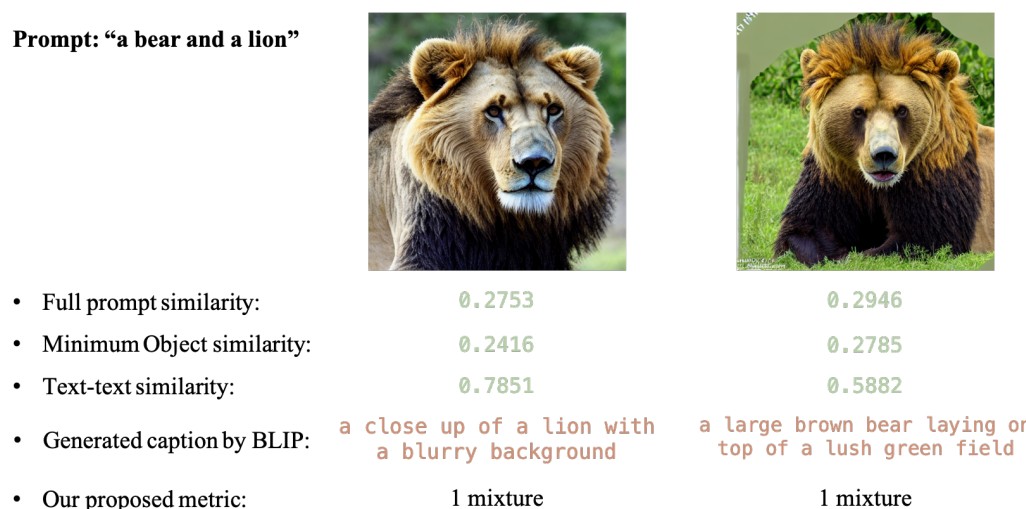

**Prompt: "a bear and a lion"**

- Full prompt similarity:            0.2753                    0.2946
- Minimum Object similarity:         0.2416                    0.2785
- Text-text similarity:              0.7851                    0.5882
- Generated caption by BLIP:    a close up of a lion with     a large brown bear laying on
                                   a blurry background           top of a lush green field
- Our proposed metric:             1 mixture                   1 mixture

Figure 9: In two images both with mixed objects, *full prompt similarity*, *minimum object similarity*, and *text-text similarity* all vary greatly, making the evaluation metrics unreliable for object mixture and missing.

### C.1 Demonstration of bounding boxes interaction corresponding to the evaluation status

Considering the working mechanism of SOTA text-to-image models, when the cross-attention maps of 2 objects respond closely during denoising, there is a high probability of generating a mixture object, as Fig. 10 shows. Consequently, within the nature of the Owl-ViT detector, these mixtures can be identified by their high overlap with high confidence.

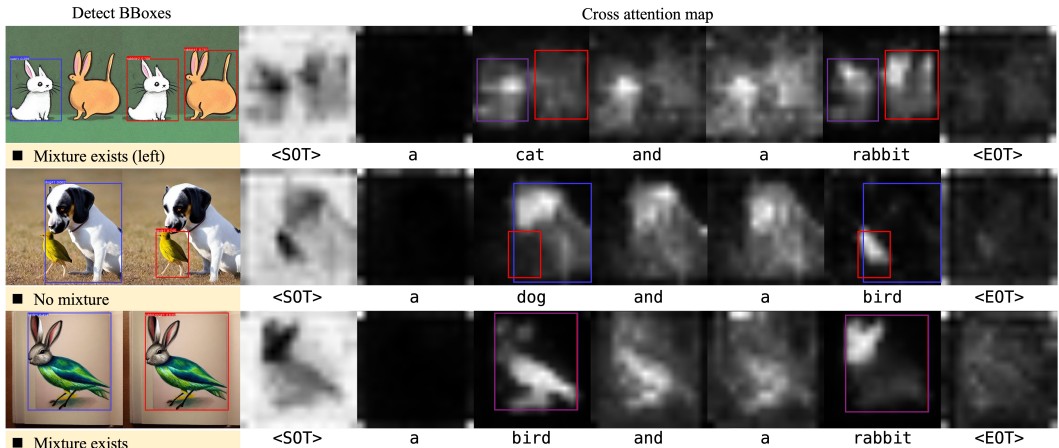

Figure 10: Demonstrating the 90% bounding box overlapping and corresponding object mixture in generated image and cross-attention maps during denoising steps.

## D Analysis of information bias with more than 2 objects

In Table 6, we demonstrate the information bias on three objects, where the problem could extend to general scenarios with a larger number of objects. In each prompt from (a) to (f), the first mentioned object always has an existence of over 20%, while the existence of the second and third mentioned objects gradually decreases to below 20% and 10%. This confirms our observation regarding information bias, which gradually overlooks the later mentioned objects. By incorporating our custom loss function $\mathcal{L}_{TEB}$, we can achieve more balanced representation of the three objects, with the existence values ranging between 10% and 20%, resulting in the information bias closer to 1.

| Prompt | (a) A/An \<obj1\> and a/an \<obj2\> and a/an \<obj3\> | | (b) A/An \<obj2\> and a/an \<obj1\> and a/an \<obj3\> | | (c) A/An \<obj3\> and a/an \<obj2\> and a/an \<obj1\> | | (d) A/An \<obj1\> and a/an \<obj3\> and a/an \<obj2\> | | (e) A/An \<obj2\> and a/an \<obj3\> and a/an \<obj1\> | | (f) A/An \<obj3\> and a/an \<obj1\> and a/an \<obj2\> | |
|---|---|---|---|---|---|---|---|---|---|---|---|---|
| | | $+\mathcal{L}_{TEB}$ | | $+\mathcal{L}_{TEB}$ | | $+\mathcal{L}_{TEB}$ | | $+\mathcal{L}_{TEB}$ | | $+\mathcal{L}_{TEB}$ | | $+\mathcal{L}_{TEB}$ |
| 3 objects exist | 0.0% | +0.8% | 0.5% | +0.0% | 0.8% | -0.3% | 0.5% | +0.0% | 0.3% | +0.0% | 1.0% | +0.0% |
| only mixture | 24.7% | -0.4% | 20.0% | +4.5% | 21.9% | +2.4% | 23.3% | +0.7% | 20.4% | +4.0% | 25.8% | -0.4% |
| obj1 + mixture | 8.6% | -2.1% | 4.3% | -0.6% | 3.0% | +1.3% | 9.5% | -3.0% | 2.8% | +0.5% | 4.5% | -1.7% |
| obj2 + mixture | 3.0% | +1.2% | 10.4% | -3.4% | 4.9% | -1.1% | 5.3% | +0.0% | 9.3% | -4.0% | 3.8% | +0.8% |
| obj3 + mixture | 4.5% | -1.3% | 3.8% | +0.5% | 11.2% | -4.1% | 3.0% | +0.7% | 5.5% | -1.8% | 8.3% | -0.1% |
| obj1 + obj2 + mixture | 1.0% | -0.3% | 0.8% | -0.5% | 0.0% | +0.3% | 1.8% | -1.0% | 2.5% | -0.3% | 0.8% | -0.5% |
| obj1 + obj3 + mixture | 1.5% | -0.5% | 0.3% | +0.2% | 2.5% | -0.4% | 1.3% | +0.0% | 0.3% | -0.3% | 1.3% | -1.0% |
| obj2 + obj3 + mixture | 0.3% | +0.0% | 2.3% | -0.3% | 3.0% | -2.2% | 0.5% | +0.7% | 0.8% | +0.0% | 1.8% | +0.5% |
| only obj1 exist | 24.7% | -5.4% | 10.1% | +1.1% | 3.8% | +5.8% | 27.3% | -5.8% | 3.3% | +6.3% | 11.0% | +0.9% |
| only obj2 exist | 14.4% | +3.1% | 29.9% | -4.6% | 12.8% | +3.6% | 6.8% | +6.0% | 30.7% | -2.8% | 4.8% | +7.2% |
| only obj3 exist | 4.8% | +6.7% | 4.8% | +4.9% | 25.1% | -5.7% | 10.3% | +1.7% | 11.1% | +2.7% | 23.8% | -3.5% |
| obj1 + obj2 exist | 7.3% | -0.8% | 8.4% | -0.9% | 0.0% | +0.0% | 3.8% | +0.2% | 5.8% | -1.5% | 0.0% | +0.3% |
| obj1 + obj3 exist | 4.8% | -0.8% | 0.3% | -0.3% | 4.4% | +0.4% | 6.8% | -0.5% | 0.0% | +0.0% | 7.3% | -0.9% |
| obj2 + obj3 exist | 0.3% | -0.3% | 4.3% | -0.8% | 6.6% | +0.3% | 0.0% | +0.3% | 7.5% | -2.8% | 6.0% | -1.4% |
| no target object | 0.0% | +0.0% | 0.0% | +0.0% | 0.0% | +0.0% | 0.0% | +0.0% | 0.0% | +0.0% | 0.0% | +0.0% |
| Info bias (obj1, obj2) | 1.72 | **1.10** | 0.34 | **0.45** | 0.30 | **0.58** | 4.04 | **1.69** | 0.11 | **0.34** | 2.32 | **1.00** |
| Info bias (obj1, obj3) | 5.16 | **1.67** | 2.11 | **1.15** | 0.15 | **0.49** | 2.66 | **1.79** | 0.30 | **0.69** | 0.46 | **0.59** |
| Info bias (obj2, obj3) | 3.00 | **1.52** | 6.21 | **2.59** | 0.51 | **0.84** | 0.66 | **1.06** | 2.77 | **2.02** | 0.20 | **0.59** |

Table 6: **Analysis of information bias in multiple objects.** When there are more objects in the prompt, the bias might gradually enlarge as the order of the mentioned objects gets farther from the first mentioned object. In this setting, we utilize the same evaluation metric to define the information bias among three objects, which is divided into pairs of (obj1, obj2), (obj1, obj3), and (obj2, obj3). *As the info bias approaches 1, it yields more balanced results regarding the existence of the object.*

## E More qualitative and quantitative comparisons in T2I-CompBench [5]

More qualitative results are in Fig. 11, Fig. 12, Fig. 13, and Fig. 14. The public benchmark, e.g., T2I-CompBench [5], contains more diverse prompts, e.g., adjective binding with noun, that would make the experiment distract. Thus, in the main paper, we follow related works, e.g., A&E (SIGGRAPH'23) [1] and Predicated diffusion (CVPR'24) [20], to create the most suitable benchmark for our task. While we respectfully disagree with the directness to conduct public T2I-CompBench to demonstrate our performance, we still conduct quantitative and qualitative comparisons in T2I-CompBench and prove our effectiveness.

### E.1 Color set

We include a quantitative comparison in Table 7 to compare SD 1.4, SD 1.5, ELLA (ArXiv'24; on the only released version, SD 1.5), SDXL-Turbo [19], and SD3 [2] on the color set within the T2I-CompBench [5] with 1,000 cases. Within all the baseline methods, our TEBOpt improves the 2 object co-existence rate with 7.9% on SD3 by addressing object mixture and object missing. Also, TEBOpt makes the generated information more balanced, where the info bias is from 2.07 to 1.30.

### E.2 Spatial set

We conducted our TEBOpt experiments using Stable Diffusion (SD) 1.4 on the set of spatial relationships (e.g., "next to," "on the side of," etc.) within the T2I-CompBench [5]. This dataset includes nouns representing 5 types of people (man, girl, etc.), 16 types of animals (giraffe, turtle, etc.), and

| Method | SD 1.4 | +$\mathcal{L}_{TEB}$ | SD 1.5 | ELLA | ELLA+$\mathcal{L}_{TEB}$ | SDXL-Turbo | +$\mathcal{L}_{TEB}$ | SD3 | +$\mathcal{L}_{TEB}$ |
|---|---|---|---|---|---|---|---|---|---|
| 2 objects exist | 25.4 % | 30.8 % | 24.5 % | 57.0 % | 63.1 % | 57.1 % | 60.5 % | 68.4 % | 76.3 % |
| only mixture | 2.9 % | 3.3 % | 3.8 % | 1.5 % | 2.6 % | 3.6 % | 2.6 % | 2.1 % | 2.3 % |
| obj1 + mixture | 1.4 % | 1.9 % | 2.4 % | 2.6 % | 0.9 % | 2.5 % | 2.4 % | 2.9 % | 2.5 % |
| obj2 + mixture | 2.5 % | 1.5 % | 1.3 % | 2.1 % | 0.6 % | 1.3 % | 1.6 % | 2.1 % | 0.9 % |
| only obj1 exist | 43.8 % | 35.7 % | 41.1 % | 20.3 % | 16.9 % | 19.8 % | 17.0 % | 15.3 % | 9.8 % |
| only obj2 exist | 19.5 % | 22.6 % | 21.3 % | 13.9 % | 13.2 % | 13.6 % | 14.9 % | 7.4 % | 7.5 % |
| no target objs | 4.5 % | 4.2 % | 5.6 % | 2.6 % | 2.7 % | 2.1 % | 1.0 % | 1.8 % | 0.7 % |
| Info bias | 2.24 | 1.58 | 1.94 | 1.46 | 1.29 | 1.46 | 1.14 | 2.07 | 1.30 |

Table 7: Quantitative comparison with SOTA methods on the color set in the T2I-CompBench. *Reference: ELLA: Equip Diffusion Models with LLM for Enhanced Semantic Alignment (ArXiv'24)*

30 types of objects (table, car, etc.), encompassing a total of 1,000 cases. Table 8 shows that our method demonstrates improvement in increasing the 2 object co-existence with 6.8% and reducing the information bias from 1.43 to 1.21. In this experiment, we further prove that when two nouns in the given prompt are from different categories, such as "a woman and a chair," resulting in a larger text embedding distance, it causes the mixture issue in text-to-image models to be concealed beneath the surface. Thus, while our main paper focuses on the task of handling only animals, it contains different challenges compared to handling more diverse objects in different categories. In our focus, we reveal and address both the mixture and missing issue. Furthermore, in the following experiment, we demonstrate that our method is effective across a more diverse set of categories.

| Method | Stable Diffusion 1.4 | +$\mathcal{L}_{TEB}$ |
|---|---|---|
| 2 objects exist | 40.4% | +6.8% |
| only mixture | 0.2% | -0.0% |
| obj1 + mixture | 0.0% | +0.1% |
| obj2 + mixture | 0.1% | -0.1% |
| only obj1 exist | 32.3% | -5.5% |
| only obj2 exist | 22.6% | -0.5% |
| no target objs | 4.4% | -0.8% |
| Info bias | 1.43 | 1.21 |

Table 8: Quantitative Comparison on the spatial set in the T2I-CompBench.

# F    Image quality evaluation

We follow SD3 [2] to conduct the image quality evaluation on Fréchet Inception Distance (FID) with CLIP L/14 image features on the generated images and the COCO 2017 val dataset [8] in 5,000 samples. The FID of (SD 1.4, SD 1.4 + TEBOpt), (SDXL-Turbo, SDXL-Turbo + TEBOpt), and (SD3, SD3 + TEBOpt) are (133.08, 133.30), (202.50, 200.71), and (143.77, 142.20), where *FIDs are higher than we usually see from text-to-image models is because the 5,000 generated sets are based on the plain prompt structure "a <objA> and a <objB>"*. This experiment proves that visual performance is mainly affected by the selected text-to-image model as the FID for the generated images w/ or w/o TEBOpt are within marginal differences in the same model. When these 3 models work with TEBOpt, only SD 1.4 gets a 0.22 increase in FID score, while SDXL-Turbo and SD3 result in a 1.79 and 1.57 decrease in FID scores. It proves that TEBOpt would improve the general image quality.

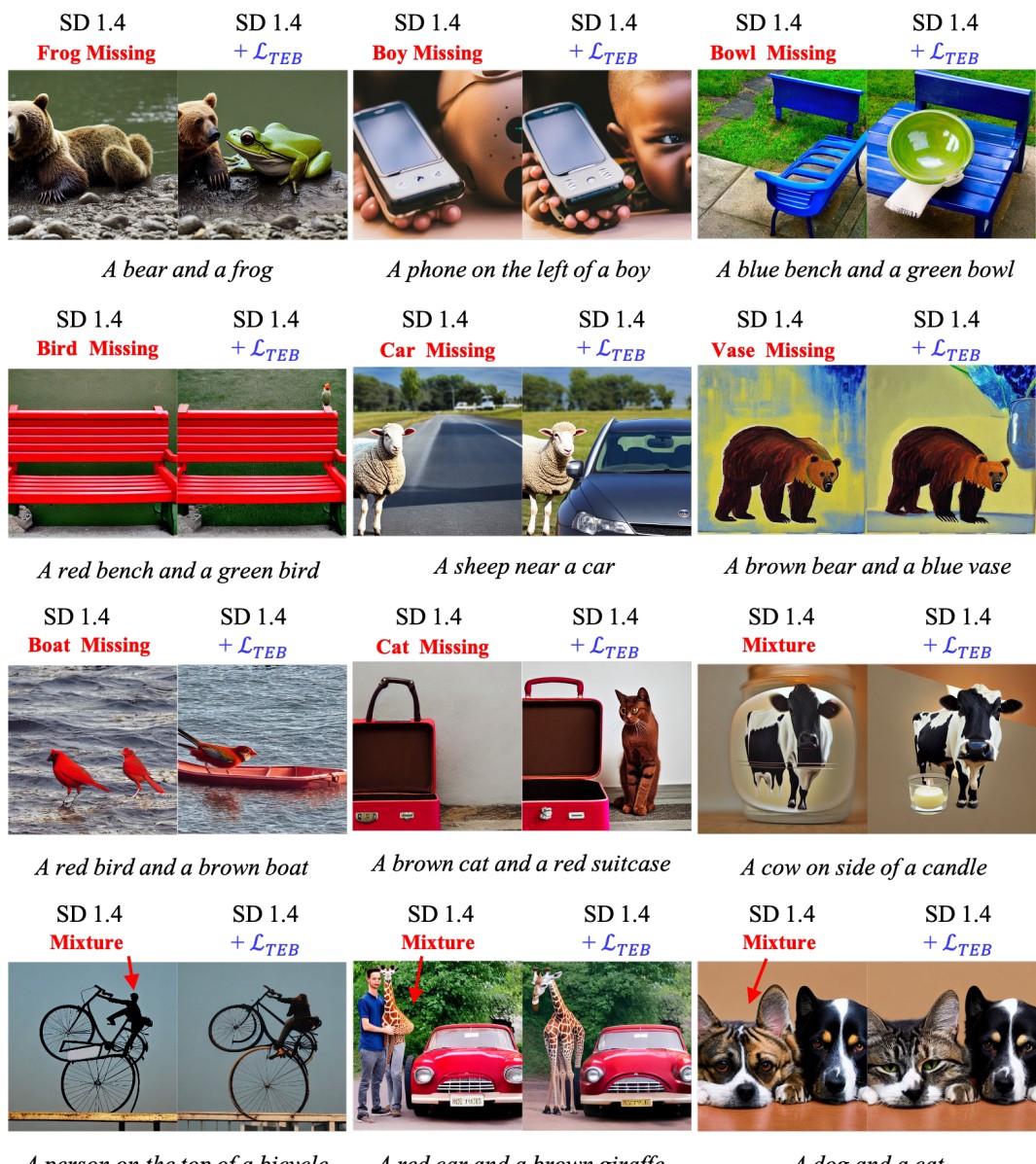

Figure 11: More qualitative results on SD 1.4 in complex prompts from color and spatial sets within T2I-CompBench [5].

## G   Interesting questions aroused by reviewers

### G.1   Would the same happen for the words may change their meaning due to nearby words? E.g., mouse?

We experimented with 1,000 samples on the effect of words that may change their meaning due to nearby words, including "mouse", "horn", "jaguar", "falcon", and "palm". Specifically, we use the prompt "an <animal/object A> and a " and evaluate the result by detecting 2 targets <animal A> and <object A> using Owl-ViT detector. Our TEBOpt can address 3.67% object missing issue in animal prompts while the optimized results may lean towards the main meaning of the word in object prompts. For example, "jaguar" tends to represent an animal rather than a car, resulting in a 1.29% decrease in generating "object jaguar" after optimization.

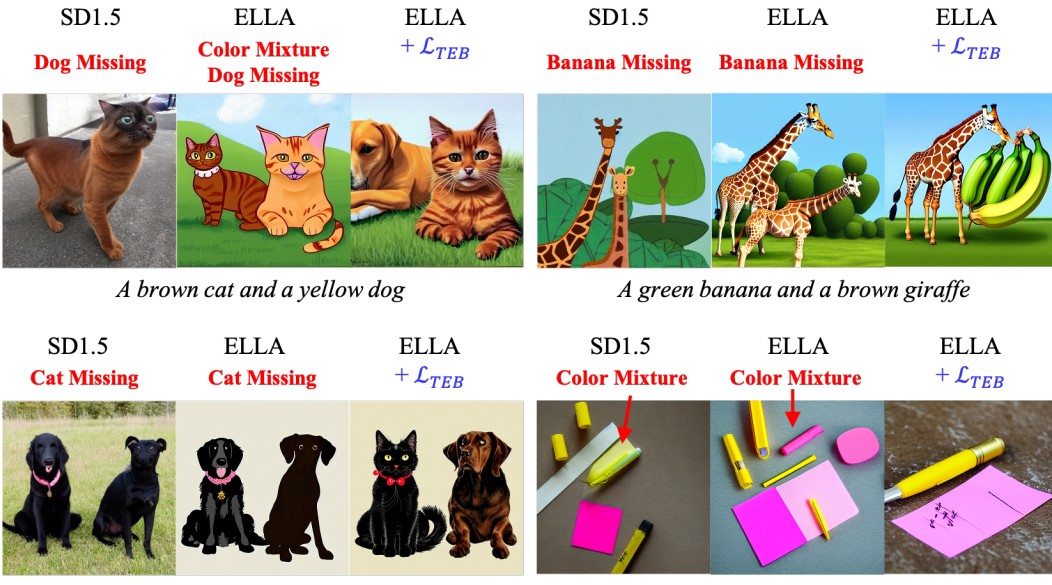

| SD1.5 | ELLA | ELLA | SD1.5 | ELLA | ELLA |

Figure 12: More qualitative results on ELLA on SD 1.5 in complex prompts from color set within T2I-CompBench [5]. *Reference: ELLA: Equip Diffusion Models with LLM for Enhanced Semantic Alignment (ArXiv'24)*

## G.2 As revealed by the analysis, not only the latter object text embedding contains the earlier object information, all the latter words all may have similar impacts to strength the text embedding of the earlier object information. Has the author considered how to resolve such possible influence?

We explored this question when we hypothesized the solution to our proposed problem. We conducted an experiment on the same 400-prompt set described in the paper, eliminating "all" earlier information from accumulating in subsequent tokens. The results are presented in Table 9. Without shared embeddings across tokens, the generation process failed to produce co-existing objects. Specifically, during the denoising process, each object token responded in the central region, as observed in the cross-attention maps, resulting in no object co-existence. In conclusion, maintaining a proper proportion of earlier object token information in the latter tokens (excluding those with concrete meanings) has more positive than negative effects, especially in generating co-existing objects within a given prompt. Therefore, we propose to optimize the critical tokens' embeddings in the paper.

| Method | SD 1.4 w/o info accumulation |
|---|---|
| 2 objects exist | 0.00% |
| only mixture | 11.75% |
| obj1 + mixture | 0.00% |
| obj2 + mixture | 0.25% |
| only obj1 exist | 46.50% |
| only obj2 exist | 39.50% |
| no target objs | 2.00% |
| Info bias | 1.18 |

Table 9: Evaluation for eliminating all information accumulation.

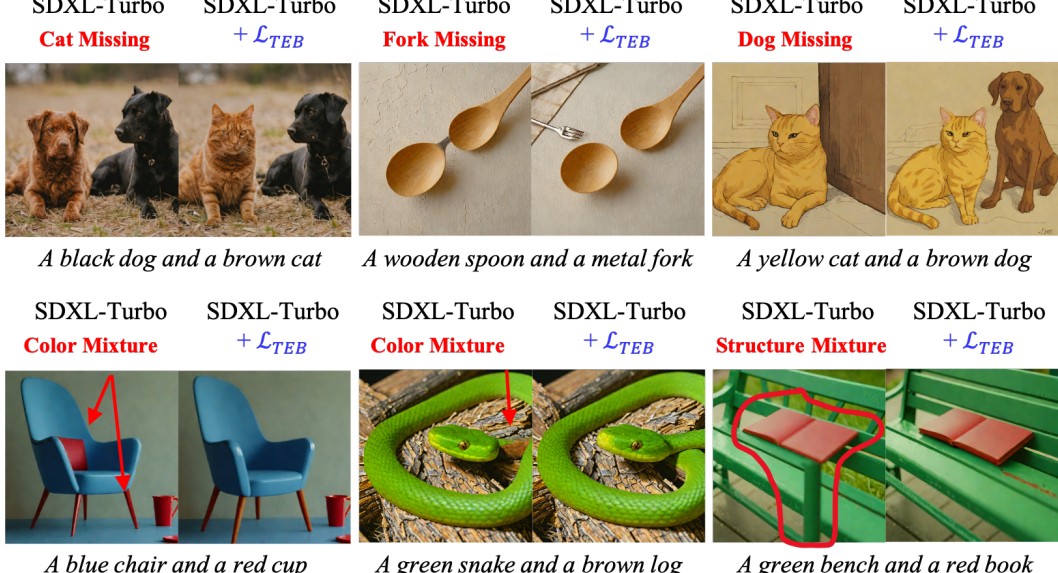

Figure 13: More qualitative results on SDXL-Turbo [19] in complex prompts from color set within T2I-CompBench [5].

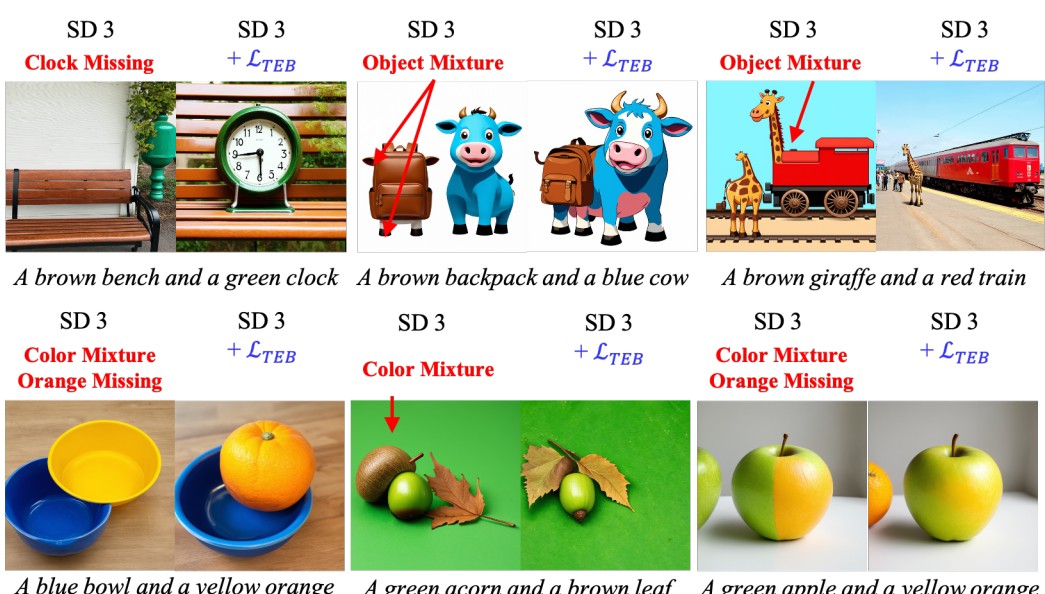

Figure 14: More qualitative results on SD3 [2] in complex prompts from color set within T2I-CompBench [5].

## H  The instruction for participants in human evaluation for our proposed evaluation metric

**Full-text instruction:**  "*You will be given an image and a text description. The text is described as "a/an <object1> and a/an <object2>". Please determine whether the objects in the image are a combination of object 1 and object 2, or if any object is missing. Select the corresponding answer from the 5 options: i) two objects exist, ii) mixture exists, iii) missing object 1, iv) missing object 2, or v) no objects exist.*" The screenshot of the human evaluation is demonstrated in Fig. 15.

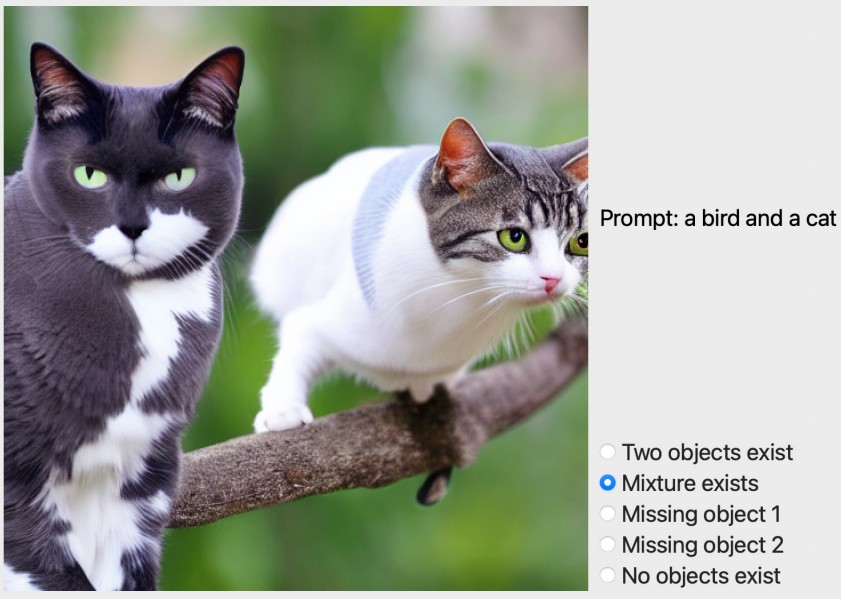

Figure 15: The screenshot of the human evaluation, containing the information and options that are given to participants.

# I Limitation

Despite the effectiveness of our proposed text embedding balance optimization in balancing information, whether the generated images contain mixed objects or lost objects is still affected by the denoising process. Furthermore, there is still room to improve in identifying the importance of information within one complex prompt. For example, this literature investigates in the prompt with equally importance objects, e.g., a <object 1> and a <object 2>, and it can be extended to more objects. In more complex prompts, such as a dog and cat playing with a mouse in front of a yard, in which the mouse is not a device, the text embedding might indicate that the device is less important.

