# OpenReview forum: "A Cat Is A Cat (Not A Dog!): Unraveling Information Mix-ups in Text-to-Image Encoders through Causal Analysis and Embedding Optimization"
_NeurIPS.cc/2024/Conference — NeurIPS 2024 poster_

### Official Review · Reviewer_fTPW · 2024-06-19

**Soundness:** 2
**Presentation:** 3
**Contribution:** 2
**Rating:** 5
**Confidence:** 4

**Summary:**

This papers address the bia problem of text-to-image generation. Unlike the previous work which focus on processing of the attention map, this work explores the text embedding, which is a key factor.  Authors conduct expensive experiments to analysis the text embedding, then propose a interesting method to address the alignment  of the text-to-image generation.  A simple text embedding alignment loss is proposed by introducing other pure  text information.

**Strengths:**

1. Authors conduct lots of interesting experiments to analysis the alignment problem, including masking the special token.

2. Exploring the text embedding instead of (stable diffusion ) UNet  network is good.

3. Authors provide the code to show the effectiveness of the presented method.

**Weaknesses:**

1. The present problem (e.g., Fig1) seems just happy the lower version of SD1. I test SD3 with the prompt : 'a lion and an elephant', the issues does not happen. In the paper, authors show the reason is from the text encoder, which I  agree. However,  the presented reason maybe does not lead to the shown challenges.

2. For text embedding, the EOT also play a key role[1]. This paper does not consider it. Only considering the soft replacing maybe harder.

3. The visual results are not good. The generated images has poor quality. Does the propose lead to other negative influence?

4. The baselines are  a little out of  fashion.  Authors should consider more recent methods: SDXL, SD3,  PlayG-v2, Ranni[2] and Ella[3].

5. Authors should use a public benchmarks to compare the proposed method, like T2I-CompBench[4].

6. The proposed idea seems limited for NeurIPS, which is not enough.

[1] Get What You Want, Not What You Don't: Image Content Suppression for Text-to-Image Diffusion Models
[2] Ranni: Taming text-to-image diffusion for accurate instruction following.
[3] Ella: Equip diffusionmodels with llm for enhanced semantic alignment.
[4] T2i-compbench: A comprehensive benchmark for open-world compositional text-to-image generation

**Questions:**

Please check the weaknesses.

**Limitations:**

Testing the presented problem in latest version of SD, like SD3.

---

> ### Author Rebuttal · Authors · 2024-08-06
>
> Dear reviewer,
>
> We appreciate your valuable suggestions and taking the time to review our work! We have carefully considered your feedback, and we would like to address your concerns. In the following section, we respond to each weakness, while weakness 3 is included in the overall rebuttal at the top of this page. Furthermore, we include corresponding quantitative and qualitative results in Table 1 and Fig. 1 in the rebuttal PDF.
>
> **Weakness 1:**
> We acknowledge that the issue occurs less frequently in SD3, as indicated in Table 1 in the rebuttal PDF. While SD3 was announced on Feb. 22, it was only open-sourced on June 12, which was after our paper was submitted. However, the problem is not completely resolved in SD3. In our experiments with the prompt "a lion and an elephant" using 1,000 random seeds, SD3 still exhibited the mixture issue in 2.9\% of cases and the missing issue in 3.6\% of cases. From SD 1 through SD 3, including SD XL, more text encoders with additional parameters have been integrated into stable diffusion models, with CLIP being a common choice. This underscores the importance of researching how the fundamental CLIP text encoder contributes to T2I models, a topic that has received little attention but is thoroughly researched in our paper.
>
> **Weakness 2:**
> Thank you for the suggestion. We analyzed the role of EOT in the Fig. 3 in main paper. In our experiment, when we masked the token embeddings for "a," "cat," "and," "a," and "dog," the remaining embeddings still generated both a dog and a cat in the image due to the information accumulated in the EOT, as discussed in line 154. EOT positively contributes to object coexistence, which aligns with [1] that discusses EOT's role in eliminating negative concepts in prompts. Therefore, we chose not to optimize EOT in our proposed method.
>
> **Weakness 4 and 5:**
> We include a quantitative comparison in the rebuttal PDF as Table 1 to compare SD 1.4, SD 1.5, ELLA [3] (on the only released version, SD 1.5), SDXL-Turbo, and SD3 in the color set in T2I-CompBench [4] with 1,000 cases. Within all the baseline methods, our TEBOpt improves the 2 object co-existence rate with 7.9\% on SD3 by addressing object mixture and object missing. Also, TEBOpt makes the generated information more balanced, where the info bias is from 2.07 to 1.30. More visual results are provided in Fig. 1 in the rebuttal PDF.
>
> **Weakness 6:**
> We carefully consider this feedback and would like to share more thoughts on the decision of the basic model and benchmark.
>
> Initially, we focused on the SD 1.4 model because it contains one commonly-used text encoder, CLIP, the research findings of which could apply to other latest models. For example, current SOTAs, e.g., SDXL and SD3, are equipped with the CLIP text encoder. Regarding the evaluation benchmark, we created our own benchmark for effectively analyzing our proposed problem because there is no suitable benchmark.
>
> To respond, we apply TEBOpt to the latest models, including SD 1.5 with ELLA, SDXL-Turbo, and SD3, on the public T2I-CompBench. We demonstrate that the proposed problem is still not addressed in the latest models, e.g., SD3. Our TEBOpt indeed addresses our proposed problem in all 3 models as well as in the more complex prompt. The visual results are in Fig. 1 in the rebuttal PDF. Specifically, on SD3, our TEBOpt increases 7.9% improvement for 2 object co-existence and makes information bias from 2.07 to 1.30.
>
> Furthermore, our paper conducts a thorough experiment on the contribution of the text encoder to text-to-image models, which is critical but has received little attention. Given the extensive discussion and rapid development of T2I models, and the emerging development of text-to-video models, we believe our task contributes significantly to the community.
>
>
>
> **References:**
>
> [1] Get What You Want, Not What You Don't: Image Content Suppression for Text-to-Image Diffusion Models. ICLR, 2024.
>
> [3] Ella: Equip diffusionmodels with llm for enhanced semantic alignment. ArXiv, 2024.
>
> [4] T2I-CompBench: A comprehensive benchmark for open-world compositional text-to-image generation. NeurIPS, 2023.

---

> ### Author Response · Authors · 2024-08-12
>
> Dear Reviewer,
>
> We appreciate your valuable suggestions and we are eager to address your concerns with the submitted rebuttal. As the author-reviewer discussion will end soon, we are looking forward to your further consideration of our submitted rebuttal. Thank you again for taking the time to foster a thorough review of our task!
>
> Sincerely,
> authors

---

> > ### Comment · Reviewer_fTPW · 2024-08-13
> > **Official comments by  Reviewer fTPW**
> >
> > Thanks authors for rebuttal. Based on the mentioned problems (e.g., old version SD, latest baselines, and public benchmark). I think this paper is not ready for NeurIPS.  For one submitted paper, since both the baseline and benchmark are not convincing,  I can not be positive.

---

> ### Author Response · Authors · 2024-08-13
>
> Dear reviewer,
>
> **old version SD:** The SD3 you suggest is released at 12 Jun., which is later than the date our task submit to NeurIPS. In the rebuttal, we still provide the experiments to prove our effectiveness that our method outperforms SD3. Please refer to the Fig. 1 and Table 1 in the global rebuttal PDF.
>
> **latest baselines:** In the initial submission, we compared with latest tasks addressing object missing, e.g., A&E (SIGGRAPH’23) [1], SynGen (NeurIPS’23) [13]. To respond to your suggestion, we further compare to Stable Diffusion XL-Turbo, Ella, and SD3. All the qualitative and quantitative results prove our effectiveness. Please refer to Fig. 1 and Table 1 in the global rebuttal PDF.
>
> **Benchmark:** the public benchmark, e.g., T2I-CompBench, contains more diverse prompts, e.g., adjective binding with noun, that would make the experiment distract. Thus, we follow related works, e.g., A&E (SIGGRAPH’23) [1] and Predicated diffusion (CVPR’24) [16], to create the most suitable benchmark for our task. While we respectfully disagree with the directness to conduct public T2I-CompBench to demonstrate our performance, we still conduct quantitative and qualitative comparisons in T2I-CompBench in the rebuttal and prove our effectiveness. Please refer to Fig. 1 and Table 1 in the global rebuttal PDF.
>
> **We sincerely look forward to more concrete details about which part of the rebuttal does not address your concern.**
>
>
> [1] Hila Chefer, Yuval Alaluf, Yael Vinker, Lior Wolf, and Daniel Cohen-Or. Attend-and-excite: Attention-based semantic guidance for text-to-image diffusion models. In SIGGRAPH, 2023
>
> [13] Royi Rassin, Eran Hirsch, Daniel Glickman, Shauli Ravfogel, Yoav Goldberg, and Gal Chechik. Linguistic binding in diffusion models: Enhancing attribute correspondence through attention map alignment. In NeurIPS, 2023.
>
> [16] Kota Sueyoshi and Takashi Matsubara. Predicated diffusion: Predicate logic-based attention guidance for text-to-image diffusion models. In CVPR, 2024.

---

> > ### Comment · Area_Chair_8fhv · 2024-08-14
> >
> > Dear Reviewer fTPW,
> >
> > After reading the authors' feedback, do you still hold your thoughts on "both the baseline and benchmark are not convincing"? What baseline and benchmark would resolve your concerns? Do you think the missing results on PlayG-v2 and Ranni were significant? Some elaborations and constructive feedback would help the authors to refute and improve their work.

---

> > > ### Comment · Area_Chair_8fhv · 2024-08-14
> > >
> > > Dear authors, do you have any comments on PlayG-v2 and Ranni? Why do you exclude them?

---

> > > > ### Author Response · Authors · 2024-08-14
> > > >
> > > > Dear AC,
> > > >
> > > > **PlayG-v2** has the same architecture as SDXL (https://huggingface.co/playgroundai/playground-v2-1024px-aesthetic). Both of them contain 2 text encoders: CLIP and OpenCLIP. Therefore, we compare SDXL-Turbo as the representative for comparison. Due to its inference efficiency, we choose SDXL-Turbo over SDXL, as SDXL requires more than 10 seconds per image on a single 3090 GPU, while SDXL-Turbo only requires less than 1 second per image.
> > > >
> > > > **Ranni** has the same text encoder as SD 1.4 and SD 1.5, which all utilize CLIP; therefore, we use SD 1.5 as a representative. Moreover, Ranni incorporates additional features such as bounding boxes, learnable color embeddings, and keypoints as conditions, focusing primarily on image editing. This differs significantly from our main focus, which is to investigate the contributions of text embedding and causal manners in text encoders to text-to-image models. Thus, comparing our model with Ranni is inappropriate because it does not address the central questions of our research.
> > > >
> > > > Overall, text-to-image models are experiencing explosive growth, with dozens of new models being released every day. We focus on comparing the most relevant models that align with our primary goal—exploring causal mechanisms and text embeddings in T2I models—rather than every available T2I model with diverse input conditions, as our main target is not image editing.

---

> > > ### Comment · Reviewer_fTPW · 2024-08-14
> > > **improving my score**
> > >
> > > Thanks Ac and authors for your effort. I have improved my score. For SD3, I reviewed a few papers for neurIPS2024  also consider SD3 for the base model instead ignoring it like this papers.

---

### Official Review · Reviewer_LGCP · 2024-07-10

**Soundness:** 3
**Presentation:** 3
**Contribution:** 3
**Rating:** 6
**Confidence:** 4

**Summary:**

The paper conducts an in-depth analysis of the impact of causal mechanisms in text encoders of text-to-image (T2I) diffusion models, which can lead to information bias and loss. The authors propose the Text Embedding Balance Optimization (TEBOpt) method, a training-free method to optimize text embeddings, resulting in a 90.05% improvement in information balance. Additionally, it presents a new metric for more accurate evaluation of information loss in generated images.

**Strengths:**

a. Originality: The paper offers a new perspective on T2I models by addressing the causal nature of text embeddings, which has been overlooked. The proposed TEBOpt represents a new approach to balancing information in text embeddings without additional training.
b. Quality: Technically sound, the paper is well-supported by comprehensive analysis and experimental results.
c. Clarity: The submission is clearly written and well-organized, making complex concepts accessible.
d. Significance: The results are important for advancing the field of T2I models, offering a demonstrable improvement in handling information bias and loss.

minors:
some relevant works should be cited and dicsussed:
Zou, Siyu, et al. "Towards Efficient Diffusion-Based Image Editing with Instant Attention Masks." Proceedings of the AAAI Conference on Artificial Intelligence. Vol. 38. No. 7. 2024.
Du, Xiaoxiong, et al. "Pixelface+: Towards controllable face generation and manipulation with text descriptions and segmentation masks." Proceedings of the 31st acm international conference on multimedia. 2023.

**Weaknesses:**

a. The paper has limited qualitative results and mentions that the analysis was conducted on a total of 400 prompts. Providing more qualitative results in the main text or appendix could better demonstrate the validity of the proposed method.
b. Some of the formulas used in the paper are not thoroughly explained, making it difficult to understand the proposed method.
c. The paper proposes a novel perspective to analyze object omissions and fusion issues in the T2I model. However, the method in the paper is rather straightforward, which to some extent may affect the quality of the generated images.

**Questions:**

a. According to the description in the text, Equation 6 is intended to encourage later tokens to be as dissimilar as possible from earlier tokens’ embedding. However, in the formula, the similarity is calculated between the object token and all tokens, with the summation range being 0 to n. This seems to contradict the goal of constraining the subsequent tokens to be dissimilar from earlier tokens?
b. In the qualitative results shown in the second row of Figure 4, it can be observed that the results using the TEBOpt method do not seem to retain the visual features of the cat very well. Does this indicate that optimizing and adjusting text embeddings is likely to cause some uncontrollable changes in semantics? Additionally, when combined with other methods, it also seems to affect the overall layout of the generated images.

**Limitations:**

The paper summarizes the limitations of the proposed method in sec6, while mentioning that the case of multiple objects as well as complex prompts can be further explored. In addition, the experiments in this paper are based on simple sentences, which can be followed up to explore whether the case with complex adjectives affects the generation effects.

---

> ### Author Rebuttal · Authors · 2024-08-06
>
> Dear reviewer,
>
> We appreciate your valuable feedback and taking the time to review our work! We have carefully considered your suggestions and we would like to address your concerns. In the following section, we respond to each weakness and question, while weaknesses a and c are included in the overall rebuttal at the top of this page. Furthermore, we include corresponding qualitative and quantitative results with more complex prompts for 1,000 samples in Fig. 1 and Table 1 in the rebuttal PDF.
>
> **Minor in Strengths:** As suggested, we will discuss the related works. Specifically, Zou et al. [A] focuses on image editing based on the cross-attention maps' similarity during denoising process while Pixelface+ [B] focuses on facial manipulation based on text description and segmentation masks.
>
> **Weakness b and Question a:** Thank you for pointing out the questions about formulas. We clarify every formula related to our analysis (Eq. 3) and proposed method (Eq. 4 to 6) to provide more detailed information.
>
> 1. Eq. 3 is the hypothesis equation, *expecting to solve the balance problem by replacing the token embedding of the later mentioned object with the corresponding pure token embedding.* Due to formatting difficulty in OpenReview, we change Eq. 3 into one line and correct one mistaken condition. $$ \varepsilon' = \varepsilon_i     \textit{     if   }  i \notin (O) \textit{ or  } i = \min (O) \textit{ else  }  \varepsilon_{p_{obj_n},5}, \tag{3} $$ where n refers to the $n^{th}$ object. Eq. 3 shows how the text embedding $\varepsilon'$ combined. Given a prompt "a dog and a cat", there are two critical tokens (<dog> and <cat>) with token indexes 2 and 5 in the critical token set $O$. Since we only replace the later mentioned token with a pure token embedding $\varepsilon_{p_{obj_n},5}$, where 5 indicates the <cat> token index in the pure prompt "a photo of a <cat>", the equation indicates to remain original token embedding when token index i is not in the set $O$ or is equal to the minimum index in the set $O$. In this case, when i is equal to 5 and its corresponding n is equal to 2, the embedding would be replaced.
>
> 2. Eq. 4, 5 and 6 are our proposed method's equations: *For rendering the correct formula, we omit the "TEB" originally on the right-bottom side for each loss.*
> $$ \mathcal{L}^{pos}(\varepsilon, \varepsilon_p) =  \underset{i \in O}{\min} sim(\varepsilon_i, \varepsilon_{p(i)}), \tag{4}$$
> $$ \mathcal{L}^{neg}(\varepsilon) = \frac{_1}{^{k(m-1)}} \underset{i \in O}{\sum} \underset{j=1; j \neq i}{\sum^{m-1}} sim(\varepsilon_i, \varepsilon_j),  \tag{5} $$
> $$ \mathcal{L} = -\mathcal{L}^{pos}+\mathcal{L}^{neg}, \tag{6} $$
> where $sim (u, v) = \frac{\mathbf{u}_i \cdot \mathbf{v}_i}{\|\mathbf{u}_i\| \|\mathbf{v}_i\|}$ and m means the effective token count.
> - Eq. 4 encourages critical token embeddings $\varepsilon_i$ from given prompt (a dog and a cat) to be more similar to pure critical token embeddings $\varepsilon_{p(i)}$ from pure prompt ("a photo of a <dog>" and "a photo of a <cat>"). We design to select one with minimum similarity to update the text embedding since token with the minimum similarity requires more efforts to purify its embedding.
>
> - Thank you for pointing out the confusion. Eq. 5 encourages critical token embeddings to be less similar to other token embeddings, which has better performance than calculating only with its earlier tokens since the current formula encourages its later tokens not to accumulate all of its information. We believe it is not a conflict with our description since similarity is a mutual result between an earlier token and a later token. Here we calculate the average as the negative loss for update. Also, we correct the equation to "j=1 to m-1" since we do not use <sot> and <eot> for optimization.
>
> - Eq. 6 is the overall loss function, where the lower the better. For achieving this goal, we give a negative sign to the positive loss since it is encouraged to have a larger value/similarity. While we give a positive sign to the negative loss since it is encouraged to have a lower value.
>
> **Question b:** Thank you for your observation. Compared to the SD 1.4 result in Fig. 4 in the main paper, applying our TEBOpt method significantly reduces the lion’s visual features on the cat, although some features might not be fully eliminated. For example, within the region above the cat's head, distinct and upright strands of hair are eliminated but it still retains some shadow.
>
> The layout of the generated images is controlled by the denoising process, which is influenced by the interaction between the text embedding and the image latent. While we use the same seed for the image latent in all methods with or without our TEBOpt, ensuring the initial image latent remains the same, the optimized text embeddings can still cause changes in the layout. When TEBOpt is combined with A&E or SynGen, it will change the layout compared to SD 1.4 method because A&E and SynGen optimize the image latent during the denoising process. Our primary goal is to address issues of object mixture and omission rather than directly controlling the layout.
>
>
> **References:**
>
> [A] Zou, Siyu, et al. "Towards Efficient Diffusion-Based Image Editing with Instant Attention Masks." Proceedings of the AAAI Conference on Artificial Intelligence. Vol. 38. No. 7. 2024.
>
> [B] Du, Xiaoxiong, et al. "Pixelface+: Towards controllable face generation and manipulation with text descriptions and segmentation masks." Proceedings of the 31st acm international conference on multimedia. 2023.

---

> ### Author Response · Authors · 2024-08-12
>
> Dear Reviewer,
>
> We appreciate your valuable suggestions and the positive feedback! We are eager to share further information about our task with you in the submitted rebuttal. As the author-reviewer discussion will end soon, we are looking forward to your further consideration of our submitted rebuttal. Thank you again for taking the time to foster a thorough review of our task!
>
> Sincerely,
> authors

---

> > ### Comment · Reviewer_LGCP · 2024-08-14
> >
> > Thank the authors for their detailed rebuttal. I keep my original rating for the acceptance of this paper.

---

> > > ### Author Response · Authors · 2024-08-14
> > >
> > > Dear Reviewer,
> > >
> > > We greatly appreciate your recognition of our work and your valuable feedback. Thank you for helping us make our formulas clearer!

---

### Official Review · Reviewer_ACAY · 2024-07-12

**Soundness:** 3
**Presentation:** 3
**Contribution:** 3
**Rating:** 7
**Confidence:** 4

**Summary:**

This paper proposes Text Embedding Balance Optimization (TEBOpt) to enhance the distinctiveness between text embeddings of equally important objects when using texts as conditions for diffusion models. It begins with an intriguing and informative experiment to investigate why a prompt like "An <object1> and an <object2>" given to a standard diffusion model, such as Stable Diffusion, often results in a higher likelihood of ignoring <object2> during image generation. The masking experiment elucidates the role of causal masking in contributing to this phenomenon.
Based on these observations and analyses, the authors propose replacing the latter embedding with a pure embedding as a baseline. They then learn to increase the distance between the embeddings of the two objects while preserving the original meaning. Both qualitative and quantitative experimental results demonstrate that the proposed method helps to reduce bias towards the first object, leading to improved generation results, even compared to existing approaches that aim to address this issue.

**Strengths:**

- The method is well-motivated by the detailed error analysis and masking experiments.
- The solution proposed by the authors are novel and interesting.
- Experimental results show that the method successfully help to improve the generation quality.
- The paper is well-organized and easy-to-follow.
- Code and models are provided to better support the reproductivity.

**Weaknesses:**

- Some of the wrapped table's layout can be improved (e.g. Table 1, Table 5)
- The data used in this work can be systematically extended and organized to form a larger and good place for community to continue conducting research on this.
- The research by far focuses on a few relatively simple scenarios, where only two or three objects in a very simple sentence pattern are presented. Explorations on more challenging scenarios would be interesting.

**Questions:**

- As revealed by the analysis, not only the latter object text embedding contains the earlier object information, all the latter words all may have similar impacts to strength the text embedding of the earlier object information. Has the author considered how to resolve such possible influence?

**Limitations:**

The limitations are potential impact are discussed in the paper.

---

> ### Author Rebuttal · Authors · 2024-08-06
>
> Dear reviewer,
>
> We sincerely appreciate your valuable suggestions and supportive feedback, as well as the time you took to review our work! We love your summary of our work, demonstrating a thorough understanding. In the following section, we respond to each weakness and question, while weakness 3 is included in the overall rebuttal at the top of this page.
>
> **Weakness 1:** Thank you for the suggestion. We will enhance the aesthetics of the table.
>
> **Weakness 2:** Thank you for the suggestion. We have already built an automatic data generator and an evaluation metric, and we plan to extend and organize the data for community use. In the supplementary materials, you can find the code for systematically extending the data in 'code/data/gen\_prompt.py', and the code for the automatic evaluation metric in 'code/eval\_metrics'.
>
> **Question 1:** Thank you for your insightful comment. We explored this question when we hypothesized the solution to our proposed problem. We conducted an experiment on the same 400-prompt set described in the paper, eliminating "all" earlier information from accumulating in subsequent tokens. The results are presented in the table below. Without shared embeddings across tokens, the generation process failed to produce co-existing objects. Specifically, during the denoising process, each object token responded in the central region, as observed in the cross-attention maps, resulting in no object co-existence. In conclusion, maintaining a proper proportion of earlier object token information in the latter tokens (excluding those with concrete meanings) has more positive than negative effects, especially in generating co-existing objects within a given prompt. Therefore, we propose to optimize the critical tokens' embeddings in the paper.
>
>
> | Method |  SD 1.4 w/o info accumulation |
> |:---------:|:--------:|
> |   2 object exist   |   0.00% |
> |  only mixture      |   11.75% |
> |   obj1 + mixture |   0.00% |
> |   obj2 + mixture |   0.25% |
> |   only obj1 exists |   46.50% |
> |   only obj2 exists |   39.50% |
>  |   no target object |   2.00% |

---

> ### Comment · Reviewer_ACAY · 2024-08-12
>
> I appreciate the authors' efforts in addressing my concerns. After reviewing the authors' rebuttal and the comments from other reviewers, I maintain my initial stance that this paper serves as a valuable starting point in this direction and offers insights into the phenomenon through error analysis and initial results. Therefore, I will keep my original rating as 'accept.'
>
> However, there is still much room for improvement. As noted by all the reviewers, the current version examines the problem in a relatively simple setting, and there are opportunities to further explore the problem setting, model choices, and evaluation benchmarks more systematically (though some have been provided in the rebuttal). I hope the authors can continue refining their work to better benefit the community.

---

> ### Author Response · Authors · 2024-08-12
>
> Dear Reviewer,
>
> We appreciate your recognition of our work as a valuable starting point! Thank you also for your thorough and thoughtful comments. We will incorporate your valuable suggestions into an updated version and continue contributing to the community.
>
> Thank you again for your support!

---

### Official Review · Reviewer_G9At · 2024-07-16

**Soundness:** 2
**Presentation:** 2
**Contribution:** 3
**Rating:** 6
**Confidence:** 5

**Summary:**

Background: Text-to-image diffusion models, such as Stable Diffusion, frequently encounter difficulties in accurately generating images from textual descriptions, especially when there are multiple objects -> object mixing and missing.

This paper focuses text embeddings to solve the problems.

Hypothesis: self-attention mixes text embeddings of multiple objects.

Analyses:
* 400 prompts of "a/an <animal 1/2> and a/an <animal 2/1>" with 17 animals produce lots of mixture of objects and missing objects
* Contribution of text embeddings to the generated images
    * Masking the dominant embeddings reduces the mixture rate
    * Replacing the embedding after the text encoder with the pure text embedding does not solve the problem.
* Process of conditions getting lost and biasing towards the first-mentioned object due to masks in self-attention in the text encoder.

Method:
* Optimize the target text embedding toward pure embedding and against earlier (first-mentioned) token’s embedding

**Strengths:**

This paper provides useful analyses on text embeddings for text-to-image diffusion models.

The experiment design corresponds to the hypothesis.

The proposed metric is a good effort for quantitative evaluation.

**Weaknesses:**

(Overall, I like the idea of this paper but it needs more work to be a solid paper.)

W1. This paper covers only nouns, especially animals. The research community has tackled different aspects of text prompts [composable, masactrl]. How about using other classes of objects to prove generality? For instance, could this method work with text prompts like 'A <cat> standing next to the <tower>' or examples from A&E such as 'A <lion> with a <crown>'? It would be valuable to see if the approach is equally effective with these more diverse classes and prompts that users might want to apply.

W2. Bboxes overlapping more than 90% not always indicate mixture.

W3. The terms should be more clearly defined.
* Information bias and object missing are confusing.
* The critical dimension?

W4. Initial latent optimization is poorly described. How can the initial latent be updated during the denoising process? I guess intermediate latents can be updated just like [self-guidance].

W5. The baseline of Figure 5 is not defined.

W6. The appendix should provide large number of examples to ensure the proposed method is free from cherry picking.

W7. Improvements in Table 4 is marginal.

W8. An important paper is missing: [composable], [custom diffusion] and [hiper]

W9. The method requires optimization for each sentence, which can be inefficient for practical use.

Misc.
    1. Placing the caption of Table 5 under the table would be better for consistency.
    2. Some numerical values in the main text are difficult to understand at first glance. To improve readability, I think it would be better to highlight or mark the key values within the table itself. For example, the main text mentions 125.42%, but I couldn't immediately identify which part in Table 4 shows a difference of 125.42%.
    3. The result shows that the balance performance in SynGen decreased by 10.65%, rather than improved.
    4. I think Figure 7 is not an appropriate example for demonstrating the inadequacy of full prompt similarity, as neither image correctly matches the given prompt 'a cat and a penguin'. The left image shows a mixture of the two animals, while the right image displays two cats instead of one.

[composable] Compositional visual generation with composable diffusion models, Liu+

[masactrl] MasaCtrl: Tuning-Free Mutual Self-Attention Control for Consistent Image Synthesis and Editing, Cao+

[self-guidance] Diffusion Self-Guidance for Controllable Image Generation, Epstein+

[custom diffusion] Multi-Concept Customization of Text-to-Image Diffusion, Kumari+

[hiper] Highly Personalized Text Embedding for Image Manipulation by Stable Diffusion, Han+

**Questions:**

Q1. Would the same happen for the words may change their meaning due to nearby words? E.g., mouse?

Q2. I do not think the self-attention layer models "causal" relationship between words. I recommend replacing "causal manner" with "dependency". Thoughts?

**Limitations:**

Some are covered.

---

> ### Author Rebuttal · Authors · 2024-08-07
>
> Dear Reviewer,
>
> Thank you for taking the time to review our work and providing us with your valuable suggestions and insightful comments. We have carefully considered your feedback, and we would like to address your concerns. In the following section, we respond to each weakness (W) and question (Q), while weakness 1 is included in the overall rebuttal at the top of this page.
>
> **W2:** We agree that 2 objects in the real-world overlapping >90% might not mean mixture, such as overlapping persons. However, in diffusion-based generated images, this high overlap does indicate mixtures due to the nature of the models used. Considering the working mechanism of SOTA text-to-image models, when the cross-attention maps of 2 objects respond closely during denoising, there is a high probability of generating a mixture object, as Fig. 2 in the rebuttal PDF shown. Consequently, within the nature of the Owl-ViT detector, these mixtures can be identified by their high overlap.
>
> **W3:**
> - Clarification for Information bias and object missing: Due to the causal masking manner, the earlier token information accumulates to all the later token embeddings, making the overall text embedding biased towards the token information mentioned earlier in the text prompt. We call this scenario information bias. Then, biased text embedding, having more information of the earlier mentioned token, makes the denoising process to have higher probability to generate images with the object mentioned earlier and leads to object missing.
> - Thank you for raising the question. We will change the term critical "dimension" to critical "token embedding". Take the prompt “a dog and a cat” as an example. There are 2 critical tokens, <dog> and <cat>, and we generate the pure text embeddings for them by the prompts "a photo of a <dog>" and "a photo of a <cat>", respectively. Then, we use the <dog>'s and <cat>'s token embeddings as critical (pure) token embeddings for calculating positive loss in TEBOpt.
>
> **W4:** Our TEBOpt optimizes the text embedding based on pure token embedding and original token embedding "before" the denoising process while the text embedding is fixed during denoising process for fixing the generated meaning guidance. Regarding the “initial” image latent mentioned in L#197, we will polish the wording to reduce confusion.
>
> **W5:** The baseline in Fig. 5 is SD 1.4.
>
> **W6:** No problem. Please refer to Fig. 1 in the rebuttal PDF.
>
> **W7:** We statistically calculated the p-value by the chi-square test in Table 4 to prove significance. The p-values for SD 1.4 vs. SD 1.4 + TEBOpt on info bias and overall performance, including 2 objects exist, object mixture, and object missing, are 0.0052 and 0.0108, which are far below the alpha value of 0.05, proving significant improvement.
>
> **W8:** As suggested, we will discuss the related works. Specifically, [composable] separates the given prompt into different objects and generates images using a set of diffusion models. [custom diffusion] focuses on preserving the customized concept for text-to-image (T2I) generation while its limitation (Fig. 9) is exactly our main focus that pretrained T2I models struggle with similar compositions, e.g., dog and cat. [Hiper] reconstructs text embedding for image editing, while its main target for image editing is different from our task.
>
> **W9:** Compared to retraining the text encoder directly, our approach is highly cost-efficient as it employs a training-free inference design, serving as a proof of concept. Specifically, our optimization adds only 3.11% to the inference time, based on an average of 1,000 cases using a single A100 GPU.
>
> **Mics (M) 1:** Thank you for the suggestion. We will make the tables consistent.
>
> **M2:** To improve numerical readability, we acknowledge that 125.42% is calculated in 2 steps. The info bias value indicates less bias when it is closer to 1, different from the intuitive assumption that lower values represent less bias. Here’s the detailed calculation:
> 1. We calculate the bias distance between the info bias value and the balanced value (which is 1). The bias distances of SD and SD + TEBOpt are (2.647-1)/1 = 164.71% and (1.393-1)/1 = 39.29%. *Note that we have rounded the values for information bias in Table 4, reporting 2.647 as 2.65 and 1.393 as 1.39.*
>
> 2. The balance improvement: 164.71%-39.29% = 125.42%.
>
> **M3:** The balance performance is assessed by how close the information bias is to 1, as this indicates a more balanced state. Thus, SynGen has a lower info bias value while SynGen + TEBOpt is indeed improved from 0.62 to 0.72 because 0.72 is closer to 1.
>
> **M4:** We considered that *full prompt similarity* is insensitive to numerical information. To describe the case more properly, we will update the image with 1 cat as in Fig. 3 in the rebuttal PDF. Still, it has a lower score than the mixture one in Fig. 7, indicating the inadequacy of the metric.
>
> **Q1:** We experimented with 1,000 samples on the effect of words that may change their meaning due to nearby words, including "mouse", "horn", "jaguar", "falcon", and "palm". Specifically, we use the prompt "an <animal/object A> and a <B>" and evaluate the result by detecting 2 targets <animal A> and <object A> using Owl-ViT detector. Our TEBOpt can address 3.67% object missing issue in animal prompts while the optimized results may lean towards the main meaning of the word in object prompts. For example, "jaguar" tends to represent an animal rather than a car, resulting in a 1.29% decrease in generating "object jaguar" after optimization.
>
> **Q2:** While we agree that "dependency" is a more accurate term, the term "causal" in this context refers to the causal masking manner used in the self-attention layer. This is described on P. 5 of the CLIP paper, where the source code function is also named "causal_attention_mask" (line 288 in modeling_clip.py). Thus, we used "causal" to maintain consistency with the terminology used in the CLIP documentation.

---

> > ### Comment · Reviewer_G9At · 2024-08-14
> >
> > I greatly appreciate the thorough rebuttal. All of my concerns are clearly resolved except below.
> >
> > In W7, I think it is marginal because there are only 0.5%p and 1%p improvement from 14.5% and 34%, respectively.
> >
> > I raised my score from 4 to 6. I anticipate an upgraded version of this paper in the camera ready.
> >
> > BTW, I value clear definition of everything. I think the word "causal" for attention mechanism is being abused considering the causal inference.

---

> ### Author Response · Authors · 2024-08-12
>
> Dear Reviewer,
>
> We appreciate your valuable suggestions and we are eager to address your concerns with the submitted rebuttal. As the author-reviewer discussion will end soon, we are looking forward to your further consideration of our submitted rebuttal. Thank you again for taking the time to foster a thorough review of our task!
>
> Sincerely,
> authors

---

### Author Rebuttal · Authors · 2024-08-06

Dear all reviewers,

We greatly appreciate the insightful suggestions and valuable comments from each reviewer. These have been immensely helpful and enlightening for refining this paper. In this section, we respond to common weaknesses and provide the PDF to include more qualitative and quantitative results on corresponding suggestions from reviewers. Enjoy!

1. **reviewer G9At (Weakness 1), reviewer ACAY (Weakness 3) & reviewer LGCP (Weakness a):** Thank you for your suggestions. We started with a plain setting to make sure the experimental findings were effective. In response, we conducted our TEBOpt experiments using Stable Diffusion (SD) 1.4 on the set of spatial relationships (e.g., "next to," "on the side of," etc.) within the T2I-CompBench [A]. This dataset includes nouns representing 5 types of people (man, girl, etc.), 16 types of animals (giraffe, turtle, etc.), and 30 types of objects (table, car, etc.), encompassing a total of 1,000 cases. The table below shows that our method demonstrates improvement in increasing the 2 object co-existence with 6.8% and reducing the information bias from 1.43 to 1.21. In this experiment, we further prove that when two nouns in the given prompt are from different categories, such as "a woman and a chair," resulting in a larger text embedding distance, it causes the mixture issue in text-to-image models to be concealed beneath the surface. Thus, while our paper focuses on the task of handling only animals, we reveal and address both the mixture and missing issue. Furthermore, we demonstrate that our method is effective across a more diverse set of categories. More qualitative results are in the Fig. 1 in the rebuttal PDF.

|          | SD 1.4 |SD 1.4 + TEBOpt |
|:--:|:--:|:--:|
|2 objects exist| 40.4% | **+6.8%** |
|only mixture | 0.2% | **-0.0%**|
|obj1+mixture | 0.0% | **+0.1%**|
|obj2+mixture | 0.1% | **-0.1%**|
|only obj1 exist | 32.3% | **-5.5%**|
|only obj2 exist | 22.6% | **-0.5%**|
|no target objs | 4.4% | **-0.8%** |
|info bias| 1.43| **1.21**|

2. **reviewer LGCP (Weakness c) & reviewer fTPW (Weakness 3):** Thank you for your thoughtful feedback. We follow SD3 [B] to conduct the image quality evaluation on Fréchet Inception Distance (FID) with CLIP L/14 image features on the generated images and the COCO 2017 val dataset [C] in 5,000 samples. The FID of (SD 1.4, SD 1.4 + TEBOpt), (SDXL-Turbo, SDXL-Turbo + TEBOpt), and (SD3, SD3 + TEBOpt) are (133.08, 133.30), (202.50, 200.71), and (143.77, 142.20), where *FIDs are higher than we usually see from text-to-image models is because the 5,000 generated sets are based on the plain prompt structure "a <objA> and a <objB>".* Visual performance is mainly affected by the selected text-to-image model as the FID for the generated images w/ or w/o TEBOpt are within marginal differences in the same model. When these 3 models work with TEBOpt, only SD 1.4 gets a 0.22 increase in FID score, while SDXL-Turbo and SD3 result in a 1.79 and 1.57 decrease in FID scores. It proves that TEBOpt tends to improve general image quality.



**References:**

[A] Huang et al. T2I-CompBench: A Comprehensive Benchmark for Open-world Compositional Text-to-image Generation. NeurIPS, 2023.

[B] Esser et al. Scaling Rectified Flow Transformers for High-Resolution Image Synthesis. ArXiv, 2024.

[C] Lin et al. Microsoft COCO: Common Objects in Context. ECCV, 2014.

---

### Author Response · Authors · 2024-08-11

Dear Reviewers,

Our eagerness for your further consideration has led us to gently remind you of the review of our submitted rebuttal. We understand that your time is valuable, and we appreciate your willingness to read our rebuttal to foster thorough consideration. We sincerely look forward to your further feedback.



Sincerely,

The authors

---

> ### Comment · Area_Chair_8fhv · 2024-08-13
>
> Dear our reviewers,
>
> Now, we are approaching the very end of the reviewer-author discussion period. Are there any further questions for the authors regarding major issues, or are your concerns sufficiently resolved? Only a limited amount of time is available for interaction with the authors.
>
> Thank you again for your dedication.
>
> Your AC

---

### Decision · Program_Chairs · 2024-09-25

**Decision:**

Accept (poster)

**Comment:**

This paper examines the role of text encoders for text-to-image (T2I) diffusion models, which can cause information bias and loss. This study analyzes text embedding's impact, especially when generating multiple objects. It introduces a training-free text embedding balance optimization method, improving information balance in stable diffusion. A new automatic evaluation metric is proposed, which measures information loss more accurately than existing methods, showing better agreements with human assessments.

We may summarize its strengths like:
- Motivation:
  - "Proposed metric is a good effort" (G9At)
  - "Well-motivated," "novel and interesting" (ACAY)
  - "Originality" and "important" results (LGCP)
  - "Exploring the text embedding instead of UNet network is good" (fTPW)
- Results:
  - "Useful analyses," "detailed error analysis"  (G9At, ACAY)
  - "Experiment design cooresponds to the hypothesis" (G9At)
  - "Technically sound" (LGCP)
- Clarity:
  - "Well-organized and easy-to-follow" (ACAY)
  - "Clearly written and well-organized" (LGCP)
- Reproductivity:
  - "Code and models are provided to better support the reproductivity" (ACAY)
  - "Authors provide the code to show the effectiveness" (fTPW)

Initially, the reviewers pointed out the following notable weaknesses:
- Applicable coverage w.r.t. class, object count, and prompts (G9At, ACAY, LGCP)
- Bad qualitative results (fTPW) and weak baselines and insufficient comparison (fTPW)

Following the author-reviewer and reviewer-AC discussions, the authors successfully resolved the major issues by providing clarifications, additional results, and a reasonable rationale for their choice of baselines. The authors' thorough responses and consistently positive feedback from the reviewers sufficiently support a recommendation for acceptance.